# Cell lineage-specific transcriptome analysis for interpreting cell fate specification of proembryos

Xuemei Zhou [1], Zhenzhen Liu [1], Kun Shen [1], Peng Zhao [1✉] & Meng-Xiang Sun[1]

In *Arabidopsis*, a zygote undergoes asymmetrical cell division that establishes the first two distinct cell types of early proembryos, apical and basal cells. However, the genome-wide transcriptional activities that guide divergence of apical and basal cell development remain unknown. Here, we present a comprehensive transcriptome analysis of apical and basal cell lineages, uncovering distinct molecular pathways during cell lineage specification. Selective deletion of inherited transcripts and specific de novo transcription contribute to the establishment of cell lineage-specific pathways for cell fate specification. Embryo-related pathways have been specifically activated in apical cell lineage since 1-cell embryo stage, but quick transcriptome remodeling toward suspensor-specific pathways are found in basal cell lineage. Furthermore, long noncoding RNAs and alternative splicing isoforms may be involved in cell lineage specification. This work also provides a valuable lineage-specific transcriptome resource to elucidate the molecular pathways for divergence of apical and basal cell lineages at genome-wide scale.

[1] State Key Laboratory of Hybrid Rice, College of Life Sciences, Wuhan University, Wuhan 430072, China. ✉email: pzhao2000@whu.edu.cn

The zygotes of model plants such as *Arabidopsis* and tobacco usually undergo asymmetric cell division to generate two daughter cells, a smaller apical cell (AC) and a larger basal cell (BC). The smaller AC divides to form the major parts of a mature embryo, the beginning of the next sporophyte generation. By contrast, the larger basal cell undergoes few cell divisions to form a suspensor composed of a few cells. Later, while the uppermost suspensor cell differentiates into the hypophysis, other suspensor cells degenerate and thus do not contribute to the next generation[1]. However, how cell fate specification of the AC lineage (ACL) and basal cell lineage (BCL) is guided by transcriptional activity is unknown[2,3].

Previously, we showed that the initial round of cell fate specification occurs at the two-cell proembryo stage[4], and the BCL possesses embryogenesis potential[5,6]. However, the transition of the BCL to the embryo also depends on developmental signals from both the ACL and maternal tissues connected to the BCL[4,6]. Therefore, ACL- and BCL-specific transcriptomes at different proembryo developmental stages may shed light on the mechanisms of their development and enable identification of the major regulatory pathways. Furthermore, to explain the different fates of the two sister cells after asymmetric division, it has been proposed that due to the polar distribution of genetic information in the mother cell after asymmetric division, some transcripts and proteins could be differentially portioned into the two sister cells. However, these mechanisms have not been reported in plant zygotes. Because single-cell proteomics is not technically feasible, a high-resolution transcriptomic analysis is needed to evaluate the above proposal.

The *Arabidopsis* embryo is a model system for cell fate determination because embryo development can be traced at single-cell resolution. The time course and cell division pattern during development of the ACL and BCL has been reported[7]. However, genomic studies of *Arabidopsis* apical and basal cells and their descendants at high resolution are hampered by the difficulty of isolating early embryos and separating the ACL and BCL. Various methods of isolating cells, such as laser capture microdissection, fluorescence-activated cell sorting, isolation of nuclei tagged in specific cell types have been used to isolate embryo proper and suspensor of globular embryos for cellular or nuclear transcriptome analysis[8–10]. These pioneer studies provided valuable data for understanding the mechanism underlying cell type establishment, but deep transcriptome studies of the apical and basal cell lineages have been challenging due to technical difficulties in isolating living early proembryos and dissecting the ACL and BCL. The high-resolution lineage-specific transcriptomes without contaminations from maternal tissues remain uncharacterized. Recently, we developed a reliable method for isolating live apical and basal cells and their descendants, which enables cell lineage-specific transcriptome profiling[11]. Here, we apply this technique to generate a transcriptional map of the ACL and BCL of early proembryos at 1-cell and 32-cell embryo stage. We demonstrate that ACL and BCL specification occurs immediately after asymmetric zygote division and identify previously unknown lineage-specific transcripts, including protein-coding genes, long noncoding RNAs (lncRNAs), and alternative splicing isoforms, associated with lineage specification. Our data provide insight into the progression of cell lineage specification during early embryogenesis.

## Results

**Construction of cell lineage-specific transcriptomes**. To investigate spatial and temporal gene expression during ACL and BCL specification in *Arabidopsis*, based on our previous study[12], we performed a comparative analysis of 2-cell proembryos (the initial stage of cell fate specification after zygote division) and 32-cell embryos (the mature stage of the BCL). The proembryos were first manually microdissected. Next, the apical and basal domains were separated for RNA sequencing (RNA-seq; Fig. 1a). To minimize the variance of the data and ensure their reliability, three independent biological replicates were performed for each cell type. Thus, 12 independent libraries were constructed and sequenced at a depth of >16 million reads per library (Supplementary Table 1). Overall, transcripts from the same cell type but different biological repeats were highly correlated ($r \geq 0.95$; Supplementary Fig. 1), indicating that the transcriptome data are suitable for analysis of cell type-specific gene expression.

After RNA-seq quality assessment, the number of expressed genes and their expression levels in each sample were analyzed. Mapping of the clean reads onto the *Arabidopsis* genome (TAIR10) resulted in detection of 13,000–15,500 genes (fragments per kilobase of transcript per million mapped reads [FPKM] > 1) in each sample (Supplementary Table 1).

Contamination of embryo transcriptome data by maternal tissues surrounding the early embryo could result in inaccurate conclusions regarding the gene-expression profile[13]. We therefore investigated the possibility of contamination in our transcriptome data sets compared to previous seed tissue transcriptome data sets using publicly available software[13]. The results revealed that the transcriptome data sets are enriched in embryos, but not in seed coats and endosperm (Supplementary Fig. 2), indicating the absence of contaminating endosperm and seed coat RNA. Thus, the high-resolution transcriptome data sets are reliable and suitable for analysis of lineage-specific transcription during cell fate specification.

**Transcriptome divergence occurs in ACL and BCL**. To investigate whether the transcriptomes are correlated with cell lineages and embryo developmental stages, RNA-seq data of isogenic embryos and from our prior study of zygotes at 24 h after pollination (Zy24)[12] were subjected to unsupervised hierarchical clustering (UHC) and principal component analysis (PCA). UHC analysis revealed cell lineage-related and developmental stage-related clustering of the ACL and BCL of early proembryos. Fifteen transcriptomes were grouped into five clusters — Zy24, AC, BC, embryo proper of 32-cell embryo (32E), and suspensor of 32-cell embryo (32S) (Fig. 1b) — further confirming that the transcriptional profile of the same cell type is consistent, whereas the profiles at different developmental stages or of different cell lineages differ significantly. PCA of the transcriptomes of the ACL and BCL across developmental stages revealed that PC1 and PC2 were sufficient to separate the samples into the five groups (Fig. 1c), consistent with the UHC results.

To further assess the transcriptional profiles, the correlation matrices of pairwise comparisons were analyzed. Consistent with the UHC result, 32S clustered away from any other cell type, including its progenitor BC (Fig. 1d), indicating a dramatic change in gene expression profile during BCL specification. Regarding the relationship between the zygote and its two daughter cells, BC displayed a closer relationship to the zygote than to AC (Fig. 1b, d), suggesting that BC and zygote have similar intrinsic gene expression profiles. For the ACL, 32E, and its progenitor AC showed a closer relationship, suggesting developmental coherence of the ACL toward embryogenesis (Fig. 1b, d). However, the temporal changes in the transcriptome of the BCL were greater than that in the ACL during proembryo development. Unexpected divergence was found between 32S and its progenitor BC. Thus, after zygote division, the AC has a distinct transcription profile while that of the BC remains similar to the zygote. However, during proembryo development,

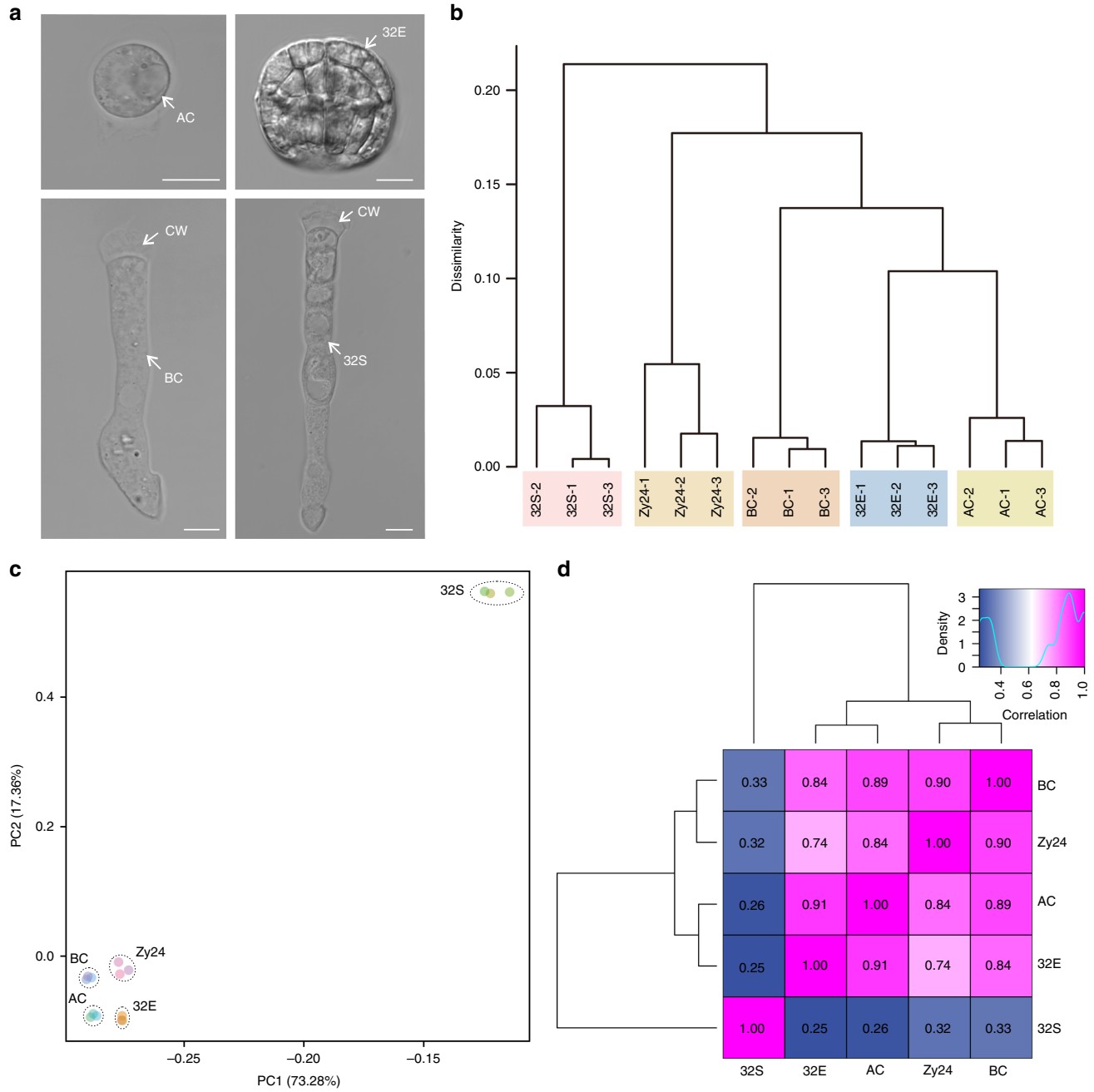

**Fig. 1 Overall analyses of the transcriptomes of zygote, apical, and basal cell lineages of early proembryos. a** Isolated living apical and basal cell lineages of early proembryos. Bar = 10 µm. **b**, **c** Hierarchical clustering (**b**) and principal component analysis (**c**) of the transcriptomes of zygote, apical, and basal cell lineages of early proembryos. **d** Pearson's correlation between the transcriptomes of zygote, apical, and basal cell lineages of early proembryos. CW cell wall, Zy24 zygote at 24 h after pollination, AC apical cell, BC basal cell, 32E embryo proper of 32-cell proembryo, 32S suspensor of 32-cell proembryo. Same abbreviations were also used in the following figures.

differentiation of the BC lineage accelerated to a highly specialized destination while the transcriptome of the AC lineage was relatively consistent.

**Temporal gene expression during ACL and BCL specification.** Hierarchical clustering of cell lineage-specific transcriptome data provides a global view of the divergence of gene expression profiles during ACL and BCL specification. To identify genes temporally regulated during ACL and BCL development, pairwise comparisons of gene expression levels between adjacent stages were performed to identify differentially expressed genes (DEGs). For the ACL, we observed significant upregulation of 3002 genes (fold change [FC]

[AC÷Zy24] ≥ 2, false-discovery rate [FDR] < 0.01) and significant downregulation of 3166 genes (FC [AC÷Zy24] ≤ 0.5, FDR < 0.01) in ACs compared with the zygote. We found that 2468 genes were upregulated in 32E compared with AC, and 2098 genes were downregulated in 32E (Fig. 2a, c and Supplementary Data 1).

In the BCL, we observed significant upregulation of 2791 genes and downregulation of 3764 genes in BCs compared with the zygote (Fig. 2b, d). Consistent with UHC analysis and PCA results, an activation wave occurred during BCL development. We found that 5124 genes were upregulated in 32S compared with in BCs, and 3943 genes were downregulated (Fig. 2b, d and Supplementary Data 1), resulting in a distinct transcriptome at

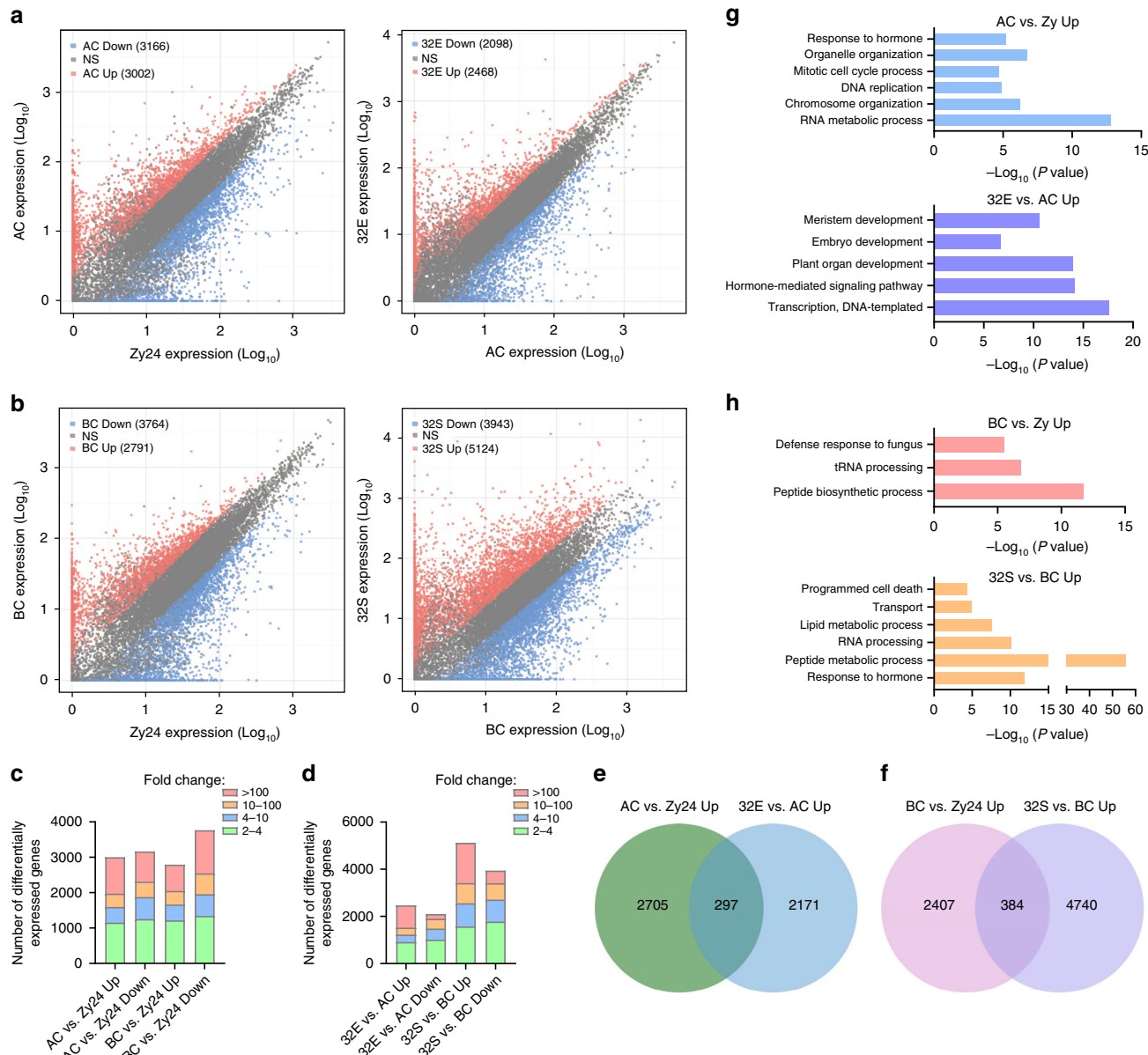

**Fig. 2 Global temporal dynamics of gene expression during apical and basal cell lineage development. a, b** Scatterplot showing temporal dynamics of gene expression in the process of apical (**a**) and basal (**b**) cell lineage specification. Pink dots indicate upregulated genes, and blue dots indicate downregulated genes. **c** Graph showing the number of differentially expressed genes between AC and Zy24 or between BC and Zy24. **d** Graph showing the number of differentially expressed genes between 32E and AC or between 32S and BC. **e, f** Comparisons of upregulated genes between AC/Zy24 and 32E/AC (**e**) or between BC/Zy24 and 32S/BC (**f**). **g, h** GO analysis of upregulated genes in apical (**g**) and basal cell linage (**h**) at 1-cell and 32-cell embryo stages, respectively.

the 32-cell embryo stage. Compared with that in the ACL much notable changes of transcriptome occurred in the BCL at the same developmental period. Interestingly, only a small portion of upregulated genes overlapped between two developmental stages in the ACL and BCL (Fig. 2e, f), suggesting developmental stage-dependent transcription during ACL and BCL specification.

Gene ontology (GO) analysis revealed that distinct molecular pathways were enriched for upregulated genes in the ACL and BCL at different stages. Upregulated genes in ACL at 1-cell embryo stage were mainly related to cell division, such as DNA replication ($P = 1.29 \times 10^{-05}$, hypergeometric test) and mitotic cell cycle ($P = 2.05 \times 10^{-05}$, hypergeometric test), whereas the genes upregulated in 32E were enriched in processes related to embryo development ($P = 2.11 \times 10^{-07}$, hypergeometric test),

plant organ development ($P = 1.03 \times 10^{-14}$, hypergeometric test), and meristem development ($P = 2.6 \times 10^{-11}$, hypergeometric test) (Fig. 2g), indicating progressive establishment of machinery for embryo pattern formation. Unlike the ACL, the genes upregulated in the BCL at 1-cell embryo stage were significantly enriched in peptide biosynthesis ($P = 2.17 \times 10^{-12}$, hypergeometric test) and defense response ($P = 3.13 \times 10^{-6}$, hypergeometric test), whereas the genes upregulated in the BCL at the 32-cell embryo stage were enriched in lipid metabolism ($P = 2.37 \times 10^{-8}$, hypergeometric test), transport ($P = 1.35 \times 10^{-5}$, hypergeometric test), and programmed cell death ($P = 3.94 \times 10^{-5}$, hypergeometric test) (Fig. 2h). Therefore, different molecular pathways are activated in the ACL and BCL for cell fate specification.

To further investigate whether the transcripts required for embryo development are preferentially expressed in the ACL after zygote division, we analyzed the expression pattern of *EMBRYO-DEFECTIVE* (*EMB*) genes during the cell lineage specification. Most known *EMB* genes[14–16] in *Arabidopsis* were coexpressed in zygotes and early proembryos (Supplementary Fig. 3 and Supplementary Data 2). These *EMB* genes were clustered into three groups based on their expression level dynamic during proembryo development. One group of *EMB* genes (group I, $n = 61$) appeared higher expression in the AC than in the BC. Notably, most *EMB* genes (group II, $n = 201$) were mainly expressed in the embryo proper at the 32-cell embryo stage, but their expression in apical and BCs appeared no significantly difference (Supplementary Fig. 3), suggesting that embryogenesis-related genes are gradually confined in the ACL in later developmental stages.

**Cell lineage-specific genes for ACL and BCL specification**. Next, spatially regulated genes between the ACL and BCL were subjected to differential gene expression analysis. At the 1-cell embryo stage, we found significant upregulation of 3454 genes (FC [AC÷BC] ≥ 2, FDR < 0.01) and downregulation of 2911 genes (FC [AC÷BC] ≤ 0.5, FDR < 0.01) in AC compared to BC (Fig. 3a and Supplementary Data 3), confirming that the gene expression profile had already diverged in the two daughter cells. As expected, we found a large increase in the number of DEGs between the ACL and BCL at the 32-cell embryo stage (4512 upregulated and 4136 downregulated) (Fig. 3a and Supplementary Data 3).

To identify lineage-specific genes that contribute to lineage specificity in 1- to 32-cell embryos, we combined the ACL and BCL results at two different stages, which yielded 1041 and 932 ACL- and BCL-maintained genes, respectively (Fig. 3b, c and Supplementary Data 4). We investigated the expression pattern of these ACL- and BCL-maintained genes during early embryogenesis. The majority of these lineage-specific genes were coexpressed in the zygote (Fig. 3b, c), suggesting them to be already transcribed in the mother cell. Also, the differences in their expression between the ACL and BCL increased as development progressed (Supplementary Fig. 4a). ACL-maintained genes were clustered into four groups based on their transcript level during zygote development and lineage specification. The genes in cluster I ($n = 370$) exhibited low expression in egg cells and zygotes, and were significantly upregulated in the ACL but their expression was low in the BCL, suggesting that the ACL-maintained genes in this cluster were due to specific de novo transcription in the ACL. Most genes in the ACL ($n = 539$, group III and IV) inherited their higher expression level from the egg cell and zygote, and were specifically downregulated in the BCL, resulting in ACL-specific expression (Fig. 3d and Supplementary Fig. 4b). Several known markers of the ACL (such as *DRN* and *AT2G24310*)[10,17] were among the ACL-ranked genes, confirming the reliability of our transcriptome data. In addition, several previously unrecognized markers (such as *ACL1* and *ACL2*) were also identified (Supplementary Data 4). The BCL-maintained genes were clustered into three groups according to their expression level during zygote development and lineage specification. Similarly, the specific expression pattern of BCL-maintained genes was due to specific de novo transcription in the BCL (Fig. 3e and Supplementary Fig. 4c; group I) or specific downregulation in the ACL (Fig. 3e and Supplementary Fig. 4c; groups II and III). The BCL-specific genes included several marker genes (e.g., *WOX8* and *PIN7*)[18,19] and unknown marker genes (such as *BCL1* and *BCL2*) (Supplementary Data 4).

To validate the transcriptome analysis results, we subjected 20 cell lineage-maintained genes to green fluorescent protein (GFP) analysis. Selected ten ACL-specific genes were coexpressed in zygotes and their expression levels differed significantly between the ACL and BCL at the 1-cell embryo stage and became restricted in the ACL at the 8-cell embryo stage (Fig. 4; Supplementary Figs. 5 and 6a). Similarly, selected ten BCL-specific genes were also coexpressed in zygotes, their expression levels differed between apical and BCs, and they gradually became restricted in the BCL as the embryo developed (Fig. 5; Supplementary Figs. 6b and 7). The GFP expression patterns of these selected twenty cell lineage-maintained genes were strongly correlated with the transcriptome data, confirming the high quality of our transcriptome dataset.

GO enrichment analysis revealed distinct roles of ACL- and BCL-specific genes in early embryogenesis (Fig. 3f, g). ACL-specific genes were enriched in embryo development- and cell division-related biological processes such as the cell cycle ($P = 1.0 \times 10^{-26}$, hypergeometric test) and chromosome organization ($P = 9.7 \times 10^{-13}$, hypergeometric test). By contrast, no embryo development-related pathways were enriched in BCL-specific genes, whereas BCL-specific genes were significantly enriched in transport ($P = 1.7 \times 10^{-9}$, hypergeometric test), fatty acid catabolism ($P = 7.2 \times 10^{-6}$, hypergeometric test), and programmed cell death ($P = 8.3 \times 10^{-4}$, hypergeometric test), in agreement with the role and cell fate of the BCL in early embryogenesis. Collectively, our results revealed that ACL- and BCL-specific genes constitute distinct molecular pathways that guide cell lineage specification during early embryogenesis.

**Cell lineage-specific lncRNAs and alternative splicing**. lncRNAs are widely distributed in animals and plants[20–23] and play important roles in the regulation of gene expression. In animals, thousands of lncRNAs have been identified in the transcriptome of embryos, and the temporary expression of these lncRNAs during embryo development were analyzed[24–26]. A series of studies on the lncRNAs revealed their critical roles in embryogenesis, such as maintaining embryonic stem cell identity, establishing the cardiovascular lineage and imprinting[27–31]. In plants, thousands of lncRNAs have also been identified in the vegetative tissues under normal and stress conditions[20,32]. Function analysis revealed that some of them were involved in photomorphogenesis[33], flowering time control[34], and male fertility[35,36]. However, the expression of lncRNAs and their roles in early embryogenesis, especially in cell lineage specification, are unclear. We detected 990 lncRNAs (of which 717 were novel) with a mean length of 1183 nt in the transcriptome data (Supplementary Data 5); these were primarily generated from the sense and antisense strands of protein-coding genes and intergenic regions (Fig. 6a–c). To validate the predicated lncRNAs, twenty of them were selected for reverse transcription polymerase chain reaction (PCR) and Sanger sequencing. The transcripts of all selected lncRNAs were successfully amplified from the cDNA prepared from 32-cell embryos (Supplementary Fig. 8). All confirmed lncRNA sequences are highly consistent with the assembled lncRNA sequences (Supplementary Data 6), confirming the high reliability of predicated lncRNAs from our cell lineage-specific transcriptomes.

Hierarchical clustering analysis of lncRNAs led to the classification of 15 transcriptomes into similar five groups as protein-coding genes, suggesting the potential roles of lncRNAs in lineage specification of early embryos (Fig. 6d). To confirm this, the expression pattern of those lncRNAs in early embryos was analyzed. Consistent with protein-coding RNAs, the lncRNAs exhibited cell lineage- and developmental stage-dependent

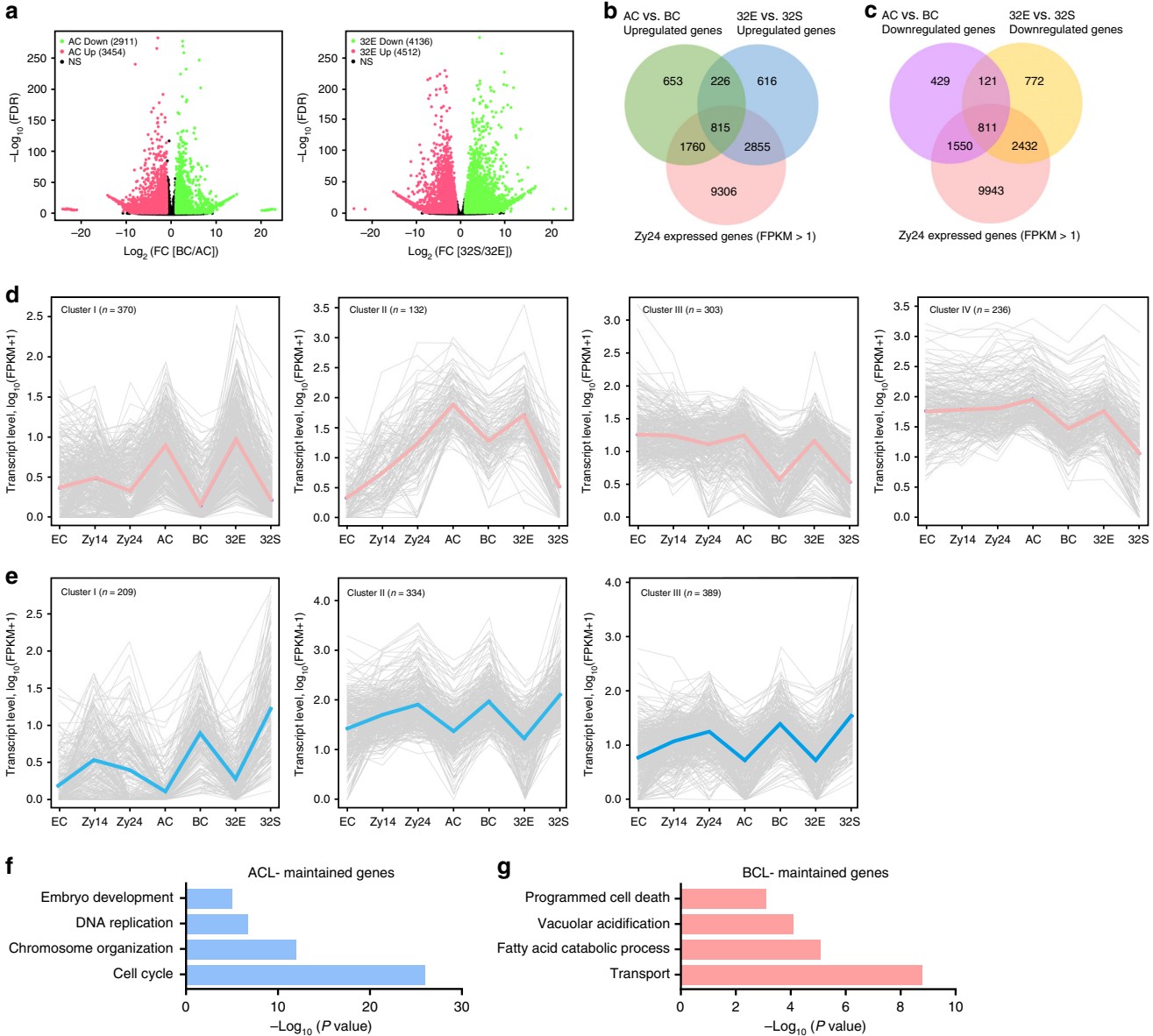

**Fig. 3 Lineage specific transcriptome analysis reveals progressive apical and basal cell lineage differentiation during early embryogenesis. a** Volcano plots display differentially expressed genes between apical and basal cell lineages at 1-cell and 32-cell embryo stages. **b**, **c** Comparisons of upregulated genes (**b**) or downregulated genes (**c**) between AC/BC and 32E/32S. **d**, **e** Expression profiles of the apical cell lineage- (**d**) and basal cell lineage-maintained (**e**) genes in egg cells, zygotes, apical, and basal cell lineages of early proembryos. Pink lines (**d**) and blue lines (**e**) indicate the mean expression level of all genes in different groups. **f**, **g** GO enrichment analysis of apical cell lineage- (**f**) and basal cell lineage-maintained (**g**) genes, respectively. EC egg cell, Zy14 zygote at 14 h after pollination.

expression. Asymmetric zygote division results in differential expression of lncRNAs in apical and BCs (Fig. 6e, f). A number of cell lineage-specific lncRNAs were identified in the ACL and BCL (Fig. 6g) and may be involved in ACL and BCL specification.

Alternative splicing, an important conserved mechanism for regulating gene expression, generates multiple transcripts from the same gene in both animals and plants. In animals, genome-wide analysis of alternative isoforms in embryos across different developmental stages have been performed and reveals that alternative splicing of pre-mRNA is developmentally regulated in embryogenesis[25,37] and associated with regulating embryonic stem cell pluripotency[38]. In plants, genome-wide analysis of alternative splicing of pre-mRNA has also been investigated during vegetative development and stress response[39–41]. However, alternative splicing in early embryo development

remains unknown. To investigate the expression of alternative transcripts in early embryos, we first examined alternative splicing events in various cells. A data set comprising 23,766 alternative splicing events of 5 different types was constructed based on our cell type-specific transcriptome (Fig. 6h). The results revealed that more than one transcript of 2500–3500 genes (18–24% of those expressed in different cell types) is expressed in different cell types (Fig. 6i, j). Notably, the alternative splicing events varied according to developmental stage and cell type. We found 1964 specific alternative transcripts in ACs and 1277 in BCs (Fig. 6k). We also identified 320 ACL-specific and 231 BCL-specific isoforms among these cell type-specific transcripts (Fig. 6k), suggesting that alternative splicing may aslo contribute to lineage specification during early embryogenesis.

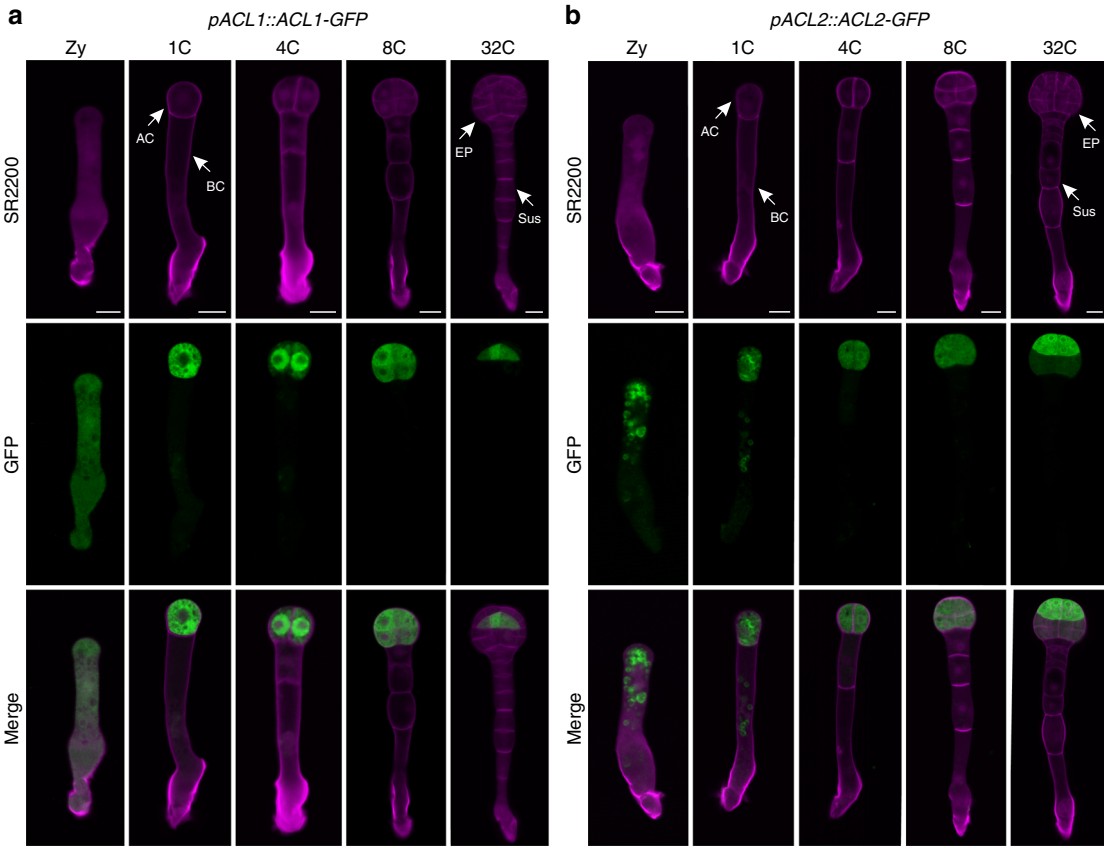

**Fig. 4 Expression pattern of selected apical cell lineage-maintained genes validates the transcriptomic analysis. a, b** Spatio-temporal expression of *ACL1* (**a**) and *ACL2* (**b**) in early embryogenesis. Zy zygote, 1C 1-cell embryo, 4C 4-cell embryo, 8C 8-cell embryo, 32C 32-cell embryo, AC apical cell, BC basal cell, EP embryo proper, Sus suspensor. Bar = 10 μm.

**Distinct transcription programs in the ACL and BCL.** To investigate transcriptional regulation during lineage specification, we evaluated the spatial and temporal gene expression pattern of transcription factor (TF) genes. The TF genes were first screened according to the annotated TF genes in the *Arabidopsis* TF database 4.0 (http://planttfdb.cbi.pku.edu.cn). We detected the expression of 954 of 1717 TF genes (FPKM ≥ 1) in both the ACL and BCL (Fig. 7a and Supplementary Data 7). Consistent with the divergence in the transcriptome, the expression of several hundred TF genes differed between AC and BC, confirming that the different transcriptional programs in AC and BC are established immediately after zygote division (Fig. 7b). We identified 73 ACL- and 39 BCL-maintained TF genes among these DEGs (Fig. 7c), which were coexpressed with other cell lineage-maintained genes in each cluster (Fig. 7d, e; Pearson's *r* > 0.87). Therefore, TFs and other lineage-specific genes form a combinatorial regulatory network for cell lineage specification during early embryogenesis.

To investigate the links between TFs and other lineage-specific genes, we screened TF-binding motifs in the promoter regions of DEGs between AC and BC or 32E and 32S (Supplementary Data 8). The binding motifs of 362 TFs were distributed in the promoter regions of these DEGs. DNA motifs corresponding to the binding sites of MYB, HD-ZIP, and Nin-like TFs were significantly enriched in the AC-upregulated genes, and bZIP and AP2 TF-binding motifs were significantly overrepresented among the BC-upregulated genes (Fig. 7f). Similarly, MYB- and ERF-binding motifs were significantly enriched in the 32E-upregulated genes, whereas bZIP-, WRKY-, and bHLH-binding motifs were significantly overrepresented in the 32S (Fig. 7g). Therefore, it is

worthy to further investigate the potential roles of these TFs and their target genes in the establishment of different transcription program in ACL and BCL for cell lineage specification (Fig. 7h, i).

## Discussion

In *Arabidopsis*, a zygote undergoes asymmetric division to generate two daughter cells with distinct sizes and developmental fates, resulting in morphologically and functionally specified ACL and BCL. The molecular mechanisms underlying apical and BC fate specification and selection of the developmental pathway are unclear. Previous studies identified genes expressed specifically in the ACL and BCL by in situ hybridization or reporter line analysis[17–19]. Attempts have also been made to profile embryo proper and suspensor cells of early proembryos using microarray technology during the past 10 years[8–10]. These pioneer works provide cell-type specific transcriptome information and valuable clues to distinguish the identities of different embryonic tissues during embryo pattern formation. However, genome-wide transcriptome analysis of apical and BC lineage specification is still hampered by technical difficulties in obtaining apical and BCs at 1-cell embryo stage. Therefore, a more complete transcriptome analysis of the ACL and BCL at different stages would shed light on the molecular mechanism of cell fate specification and facilitate identification of important regulatory factors. By overcoming technical limitations, high-resolution transcriptomes of ACL and BCL at 1-cell and 32-cell embryo stages were generated through RNA-seq technology, which fill in the gap of transcriptome dynamic during early embryogenesis and is required for the investigation of genome-wide gene activity that guides apical and BC lineage specification. Furthermore, the transcriptome data

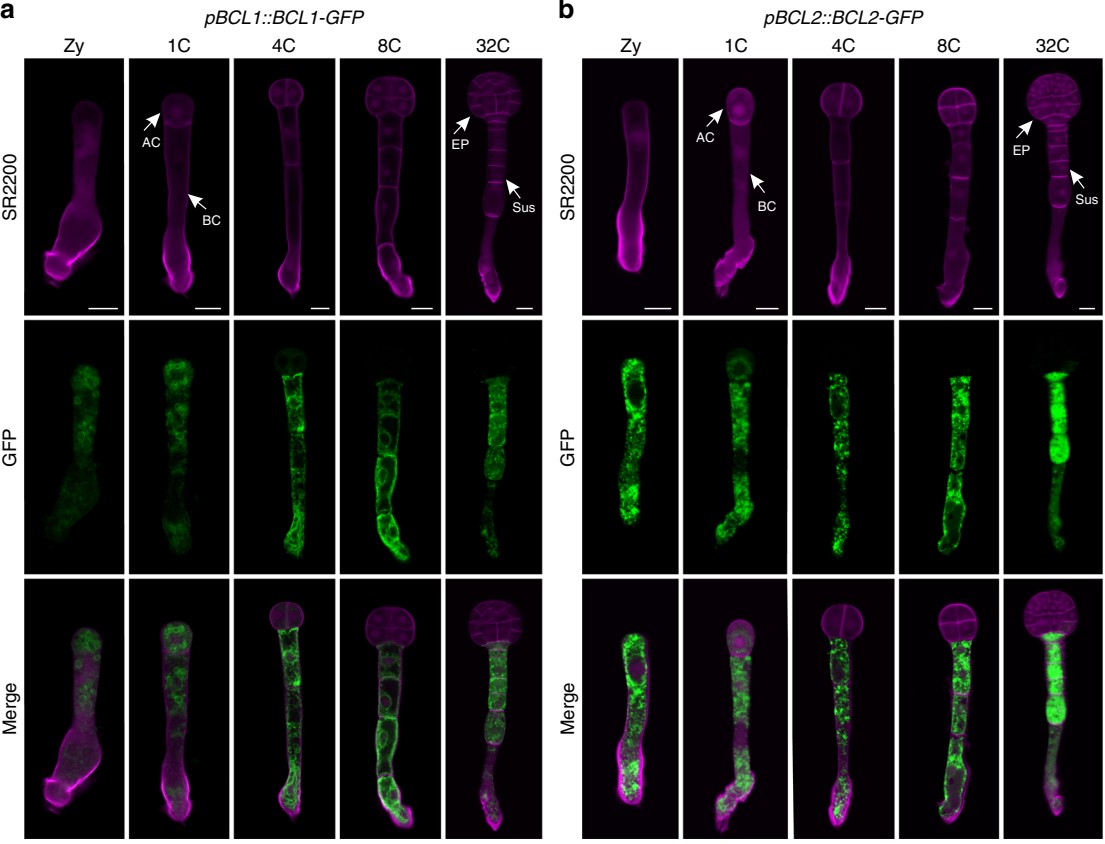

**Fig. 5 Expression pattern of selected basal cell lineage-maintained genes validates the transcriptomic analysis. a, b** Spatio-temporal expression of *BCL1* (**a**) and *BCL2* (**b**) in early embryogenesis. Zy zygote, 1C 1-cell embryo, 4C 4-cell embryo, 8C 8-cell embryo, 32C 32-cell embryo, AC apical cell, BC basal cell, EP embryo proper, Sus suspensor. Bar = 10 μm.

reported here provide more completed information on the cell type-specific gene expression which enables a comparative analysis of the data with previous available data to understand gene expression in different cell types or by different techniques (Supplementary Fig. 9).

We performed a detailed transcriptomic analysis of the ACL and BCL and identified several thousand DEGs in the two daughter cells of a zygote (Fig. 3a and Supplementary Fig. 10a–c), suggesting that the first zygote asymmetric division leads to an uneven distribution of transcripts. This is in line with our previous report that the initial round of cell fate specification occurs at the 2-cell proembryo stage[4]. Three mechanisms of the asymmetric distribution of apical and BCs have been proposed[19,42,43]: (i) asymmetric portioning of zygote-deposited transcripts into apical and BCs, (ii) specific de novo transcription in apical or BCs, and (iii) selective turnover of transcripts in apical or BCs. Consistent with the first proposal, we identified 170 DEGs between apical and BCs that were highly expressed in zygotes, but at negligible levels in 32-cell embryos, suggesting that these transcripts are inherited unevenly by the two daughter cells following asymmetric zygote division (Supplementary Fig. 10d, h and Supplementary Data 9). However, 2577 DEGs between apical and BCs were due to specific de novo transcription, implying that the ACL and BCL require cell lineage-specific transcription machinery to guide cell fate specification (Supplementary Fig. 10b, c). Consistent with our results, genes specifically activated in the suspensor, such as G564 in *Phaseolus coccineus* has been identified[44]. In addition, selective transcript degradation contributes to the divergence of the ACL and BCL. These genes

are abundant in zygotes and specifically expressed in the ACL or the BCL (Fig. 3d, e and Supplementary Fig. 10f, j). We propose that asymmetric zygote division leads to uneven inheritance of transcripts; however, the distinct transcript profiles of the ACL and BCL mainly result from cell type-specific de novo transcription and selective deletion of the inherited transcripts (Supplementary Fig. 10). This establishes a cell lineage-specific transcription program to promote ACL and BCL specification during early embryogenesis.

Although it is well-known that the developmental pathways of the two daughter cells of a zygote are distinct, the molecular basis is unclear. The MAPK signaling cascade[45,46] and WRKY2-WOX8 transcription cascade[47,48] are involved in the cell fate determination of the BCL. However, the key components in regulation of cell fate determination and the differences of regulatory pathways between apical and BC lineages are unknown. A comparative transcriptome analysis revealed that transcriptome reconstruction occurs shortly after zygote division (Fig. 1), leading to different gene expression profiles between apical and BCs. Dramatic changes in gene-expression profile were found during BCL compared with ACL development (Fig. 3), suggesting involvement of different regulatory machineries. In animals, both TFs and lncRNAs regulate gene expression and cell lineage specification. For example, lncRNAs[30,49] and TFs[50–52] with potentially critical roles in lineage specification have been identified in a variety of cell types. Indeed, we found that cell lineage-specific TFs are coexpressed with other lineage-specific genes to form a regulatory network for cell lineage specification (Fig. 7). More interestingly, cell lineage-specific lncRNAs (Fig. 6) were identified

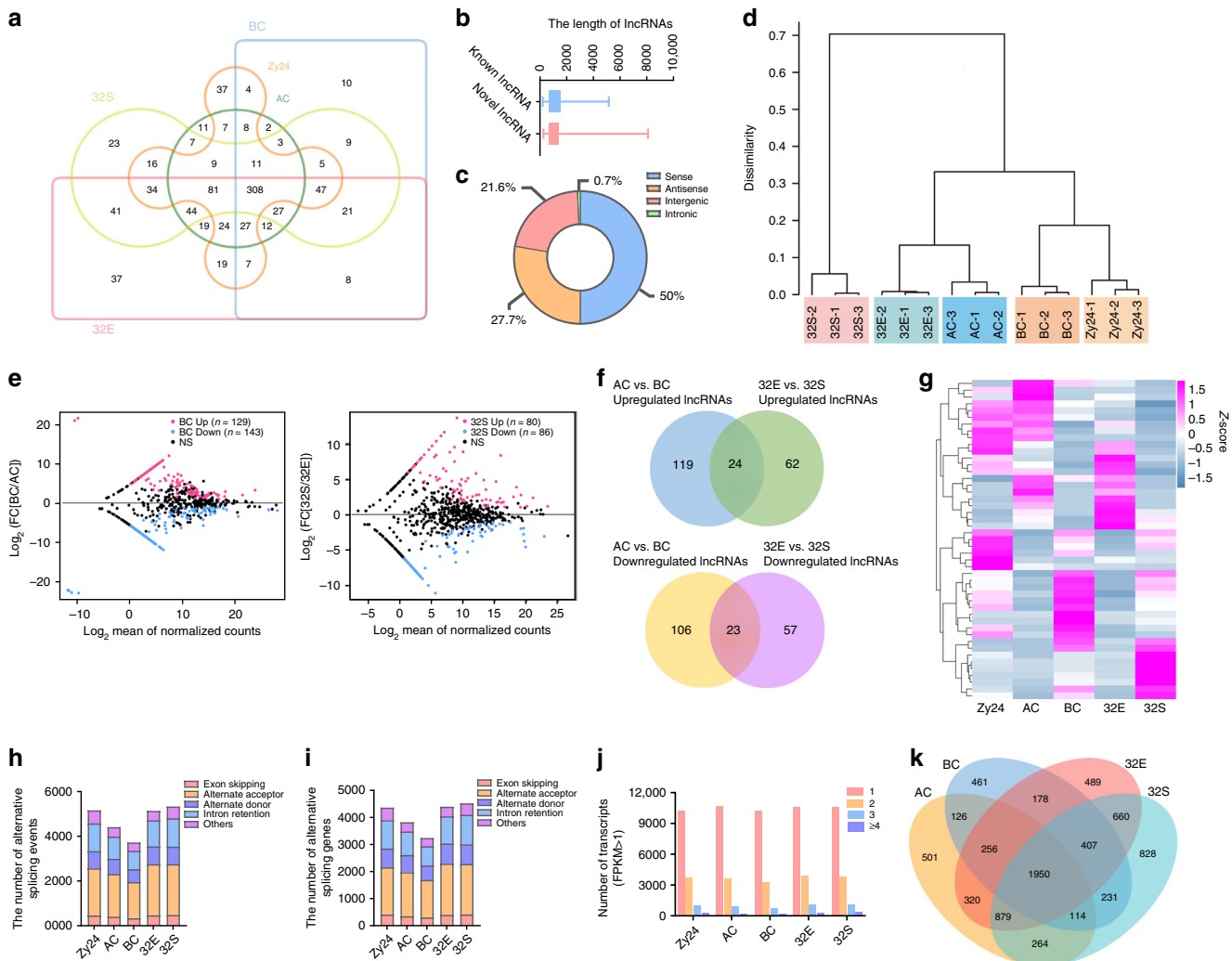

**Fig. 6 The expression of lncRNAs and alternative splicing isoforms during the process of apical and basal cell lineage specification. a** Venn diagram of the lncRNAs in five different cell types. **b** The length of known and novel lncRNAs identified in early embryos. Whiskers in box plots indicate the minimum and maximum length of lncRNAs. **c** Proportions of different types of lncRNAs identified in early embryos. **d** Hierarchical clustering of zygote, apical, and basal cell lineages of early proembryos based on the expression of lncRNAs. **e** MA plots depict differentially expressed lncRNAs between apical and basal cell, or between 32E and 32S. **f** Overlap analysis of upregulated or downregulated lncRNAs between AC/BC and 32E/32S. **g** Heatmap showing the expression of apical cell lineage- and basal cell lineage-maintained lncRNAs in early embryogenesis. **h, i** The number of alternative splicing events (**h**) and genes (**i**) in different cells. **j** The number of genes with different transcripts in each cell type. **k** Overlap analysis of alternative splicing transcripts identified in apical and basal cell lineages.

in the cell lineage-specific transcriptomes. The roles of these lineage-specific TFs and lncRNAs require investigation. Furthermore, we identified 320 ACL-specific and 231 BCL-specific isoforms among the cell type-specific transcripts, suggesting a role for alternative splicing in ACL and BCL specification (Fig. 6k). Thus, multiple factors may regulate ACL and BCL specification in a coordinated manner.

The life cycle of animals and plants typically alternates between diploid and haploid generation. Diploid generation is initiated by the fusion of a sperm cell and egg cell, giving rise to a fertilized egg cell (also called a zygote). Hence, the zygote is the starting point of early embryogenesis. However, unlike animals, in which the zygote develops into the embryo, plant zygotes undergo asymmetric division to generate two daughter cells, which make different contributions to the new generation. The smaller daughter cell (AC) divides to form the major parts of a mature embryo, whereas the larger daughter cell (BC) contributes only to

the formation of the hypophysis; other suspensor cells degenerate and do not contribute to the next generation[1,53]. Our transcriptome analysis revealed that distinct gene expression profiles were established immediately following zygotic division. Interestingly, an unsupervised clustering analysis showed that BCs are more similar to zygotes than ACs in terms of their intrinsic gene expression profile. Consistent with these results, the BCL is reportedly capable of cell-fate conversion of the BC to a zygote-like cell, which could divide to form a typical embryo with an embryo proper and a suspensor similar to the original zygote[5]. However, when the BC is destroyed at the 1-cell embryo stage, the AC cannot return to a zygote-like state and develop into a typical embryo[5].

All these findings raise a very interesting question that at which stage embryogenesis pathway is turned on or initiated, although it seems generally accepted that the embryogenesis is triggered in zygote stage. Based on our analysis, we found no strong evidence

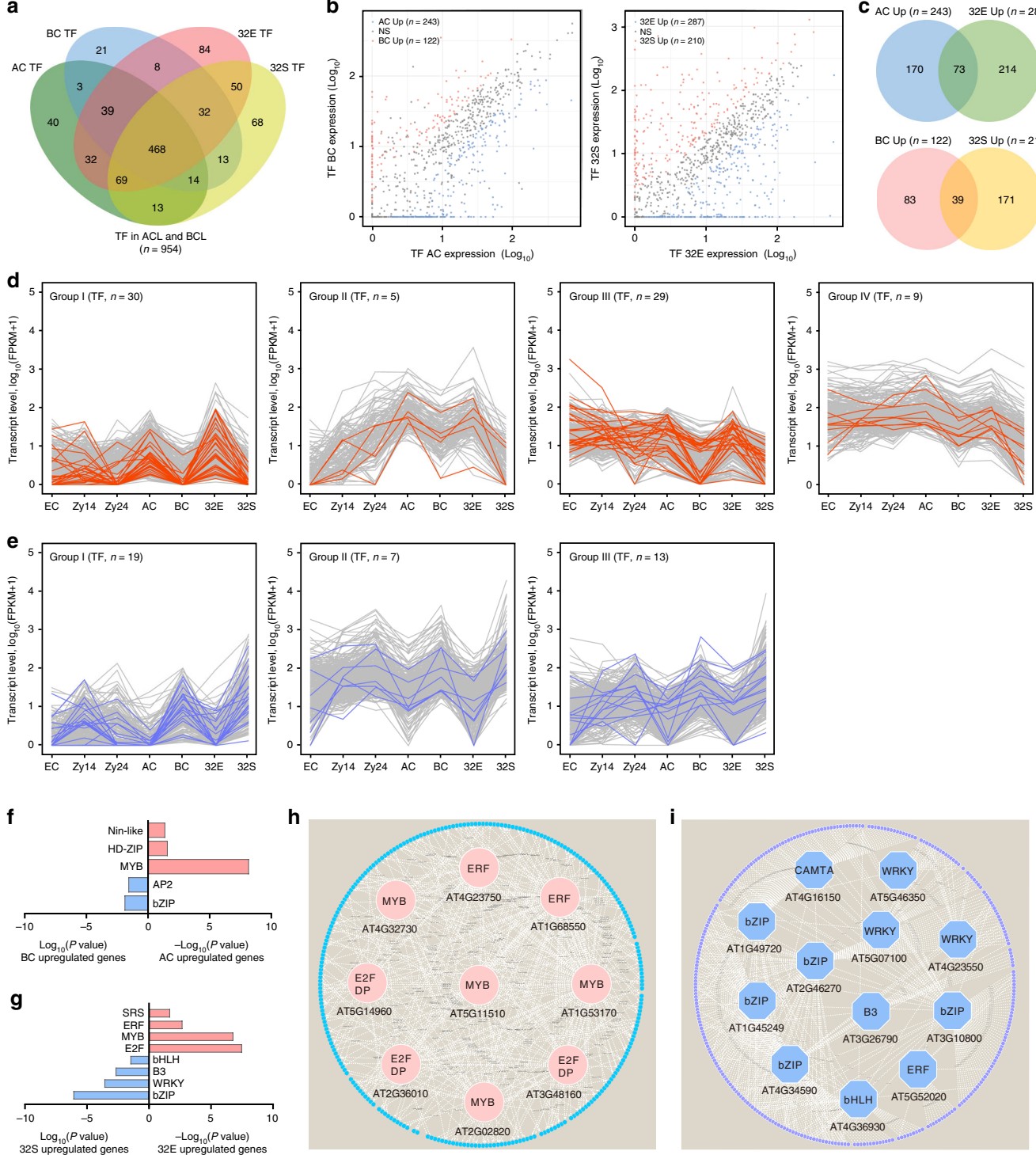

**Fig. 7 The regulatory networks of diverse transcription factors in apical and basal cell lineages of early proembryos. a** Venn diagram showing expressed TF genes in AC, BC, 32E, and 32S (FPKM ≥ 1). **b** Scatterplot showing differentially expressed TF genes between AC and BC, and between 32E and 32S. **c** Overlap analysis of upregulated or downregulated TF genes between AC/BC and 32E/32S. **d, e** Identification of TF genes that are coexpressed with apical cell lineage- (**d**) and basal cell lineage-maintained (**e**) genes in early embryogenesis. Orange lines (**d**) and purple lines (**e**) indicate the TF genes in each group. **f, g** Enrichment of TF DNA binding motifs within the promoter regions of upregulated and downregulated genes between AC/BC (**f**) and 32E/32S (**g**). **h, i** Predicated regulatory networks between enriched TFs and their potential targeted genes in apical (**h**) and basal cell (**i**) lineages.

for the conclusion, instead, we propose two possibilities for the initiation stage of embryogenesis. First, embryogenesis is triggered in zygotes and both the apical and BC inherit the necessary components for triggering embryogenesis from the zygote, which endues both cells the embryogenesis potential. However, the

embryogenesis potential of the BC is normally suppressed by the AC[6]; thus, only the AC undergoes embryogenesis. Second, embryogenesis is initiated in ACs in plants. In this case, embryogenesis is activated only in the AC by de novo transcribed factors based on the molecular components from the zygote. This

may explain why when the AC is laser ablated, the BC acts as a zygote and divides once to produce a new AC, which further divides to form an embryo[5]. Both of these proposals warrant further investigations. Further study to confirm these proposals is necessary for our research strategy in understanding the molecular mechanism regulating embryogenesis initiation. In the former case, we should search for the critical factors that trigger embryogenesis in zygotes and the factors may be shared by both apical and BCs, whereas in the latter case, the embryogenesis is triggered only in the AC and thus those factors guiding the cell to the embryogenesis pathway should be AC-specific. Thus, the comparative transcriptome analysis of apical and BC will provide valuable clues and the AC, but not the zygote, will be the target cell for our further investigation.

## Methods

**Materials**. *Arabidopsis thaliana* ecotype Columbia-0 seeds were cultivated in greenhouse under long-day conditions (16 h light/8 h dark) at $22 \pm 1$ °C.

**Embryo isolation**. Early *Arabidopsis* embryo isolation and the separation of apical and BC lineages of early proembryos were performed according to our previous protocol[11]. For embryo isolation, *Arabidopsis* seeds were collected in the enzyme solution (0.1% cellulase R10, 0.08% macerozyme R10, 80 mM D-Sorbitol, 10% glycerol, 0.058% MES, pH = 5.8) for 30 min at 25 °C, followed by three washes in the washing solution (80 mM D-Sorbitol, 10% glycerol, 0.058% MES, pH = 5.8). Early embryos were then isolated manually from the seeds with two fine glass needles under an inverted microscope. Apical and BC lineages of early proembryos were separated using LMD System (Leica, Germany), and collected under an inverted microscope by the micromanipulation. Isolated apical and BC lineages were extensively washed four times and transferred into lysis buffer (Life Technologies, USA) and stored in liquid nitrogen for mRNA isolation.

**cDNA preparation and library construction for RNA-seq**. mRNA was extracted from all sample (17–23 apical or BC cells [and their descendants] per sample) using a Dynabeads mRNA DIRECT Micro Kit (Life Technologies, USA). cDNA synthesis and double-stranded cDNA amplification were performed according to our previous protocol using a SMARTer Ultra Low RNA Kit for Illumina Sequencing (Clontech, USA). PCR-amplified cDNA was purified using an Agencourt AMPure PCR Purification Kit (Beckman Coulter). Purified cDNA was validated using an Agilent 2100 Bioanalyzer and Agilent's High Sensitivity DNA Kit (Agilent Technologies).

RNA-Seq libraries were prepared using a NEBNext Ultra DNA Library Prep Kit for Illumina (New England Biolabs) according to the reference guide, and sequenced on an Illumina HiSeq X Ten, generating about 6 Gbp raw sequence data for each library.

**RNA-Seq data analysis**. Reads that contained adapters and low quality reads (ambiguous bases >10%) were removed from the raw data using Cutadapt version 1.15 (ref. [54]) with the parameters "--trim-n -m 148 -g -G -o -p" and an in house script. Paired-end clean reads of each sample were then mapped to the genome (Col-0 genome reference, TAIR10) using Bowtie 2 (ref. [55]) with the parameters "-q --sensitive --dpad 0 --gbar 99999999 --mp 1,1 --np 1 --score-min L,0,-0.1 -I 1 -X 1000 --no-mixed --no-discordant -p 2 -k 200 -x -1 -2". Gene expression levels were quantified and normalized as FPKM measurements. FPKM value calculation was performed using RSEM[56] with the parameters "--paired-end --bam". DEGs between different cell types were identified using DESeq2 (ref. [57]) with the parameters "-list -diff -group -log₂ 1 -padj 0.01 -outdir". Genes with a fold change value $\geq 2$ and FDR < 0.01 were considered as significant DEGs. The level of contamination in the transcriptome data generated from seed coat and endosperm was assessed using the software with the default parameters (https://github.com/Gregor-Mendel-Institute/tissue-enrichment-test)[13].

**lncRNA analysis**. We identified lncRNAs from our transcriptome data according to the following procedures. Firstly, clean reads of each sample were mapped to the genome using Tophat2 (v2.1.1; http://tophat.cbcb.umd.edu/) with the parameters "-G -p 6 --library-type fr-unstranded -i 20 -I 500000 -o", and then putative transcripts were assembled using Cufflinks (v2.2.1; http://cufflinks.cbcb.umd.edu/) with the parameters "-F 0.3 -u -g -o". Second, all putative transcripts were merged using Cuffmerge with the parameters "-g -s -o -p 12" to generate a set of transcripts, and transcripts that overlapped with annotated protein-coding genes or with one exon (length < 200 nt) were removed. Third, lncRNAs were determined using following five independent programs: CNCI (v2) with the parameters "-p 1 -m pl -f -o", CPC (v0.9-r2) with the default parameters, PLEK (v1.2) with the default parameters, CPAT (v1.2.2) with the default parameters and pfam_scan (v1.6-1) with the parameters "-translate all -fasta -dir -outfile". Transcripts that

passed through at least four programs were annotated as lncRNAs. The expression levels of lncRNAs were quantified using RSEM[56] with the parameters "--paired-end --bam". Differentially expressed lncRNAs were identified using DESeq2 (ref. [57]) with the parameters "-list -diff -group -log₂ 1 -padj 0.01 -outdir".

**Alternative splicing analysis**. Alterative splicing events were identified from the transcripts (FPKM ≥ 1) assembled from the transcriptome of each cell type using ASTALAVISTA (v3.2)[58] with the parameters "-t asta -i -o", and alterative splicing events in different cell types were compared using an in house script (https://github.com/frasergen-rna/AS-cmp). Alterative splicing events were classified into four basic types including exon skipping, alternative acceptor site, alternative donor site, and intron retention using ASTALAVISTA.

**TF analysis**. The 1717 *Arabidopsis* TFs and their functional binding sites were downloaded from the plant TF database PlantTFDB (http://planttfdb.cbi.pku.edu.cn/). The functional binding sites of each TF were screened in DEGs between apical and BC lineage at different stages using FunTFBS (http://planttfdb.cbi.pku.edu.cn/). The location of each TF-binding site was confirmed using an in-house script (https://github.com/frasergen-rna/TF_prediction). Genes with TF-binding sites in the promoter regions (−500 ~ +100) and with similar expression pattern with targeted genes were chosen for further analysis. Enrichment *P* value of each TF was calculated using hypergeometric test in the R package and was adjusted using the FDR method. Predicated regulatory networks between enriched TFs and their potential targeted genes were visualized using Cytoscape (v3.7.2).

**GO analysis**. GO terms for *Arabidopsis* genes were downloaded from the TAIR (https://www.arabidopsis.org/). The number of genes in each GO term was calculated, and then GO enrichment analysis was performed using hypergeometric test in R package.

**Vector construction and plant transformation**. To generate GFP fusion protein reporter lines, DNA fragments containing promoter region and coding sequences were amplified from *Arabidopsis* genomic DNA and inserted into the vector pART 27 upstream of *GFP*. All GFP fusion vectors were transformed into *Arabidopsis* using *Agrobacterium tumefaciens* strain GV3101 according to a previous protocol[59]. The primers for DNA amplification were listed in Supplementary Table 2.

**Confocal microscopy analysis**. Isolated zygotes and embryos were stained in the solution containing 0.1% (v/v) SR2200 and 4% para-formaldehyde. GFP and SR2200 fluorescence of zygotes, and early embryos were observed using a confocal microscope (Leica TCS SP8, Germany).

**Reporting summary**. Further information on research design is available in the Nature Research Reporting Summary linked to this article.

## Data availability

Data supporting the findings of this study are available within the main text and its Supplementary Information files. A reporting summary for this Article is available as a Supplementary Information file. All RNA-seq data have been uploaded to the NCBI Gene Expression Omnibus (GEO) under accession GSE135422. The source data underlying Figs. 2, 3, 6a–c, e, f, h–j, 7 as well as Supplementary Figs. 2, 3b, 4a–c, 8, and 10d–g, h–k are provided as a Source Data file. All other data that support the findings of this study are available from the corresponding author upon request.

## Code availability

The code and software sources from previously published paper are listed in the Methods. The in house scripts for alternative splicing analysis (https://github.com/frasergen-rna/AS-cmp) and TF analysis (https://github.com/frasergen-rna/TF_prediction) are publically available from GitHub. All other details of the code used in this study are available from the corresponding author upon request.

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

## Acknowledgements

This work was supported by National Natural Science Foundation of China (31970340 and 31770355). We thank Yanbing Cheng and Gaigai Yuan (Frasergen Bioinformatics Co., Ltd.) for suggestions for data analysis.

## Author contributions

P.Z. and M.S. designed the research. X.Z., Z.L., K.S., and P.Z. performed the experiments. P.Z. and X.Z. analyzed the data. P.Z. and M.S. wrote the paper.

## Competing interests

The authors declare no competing interests.
