## [Peer Review File · Nature Communications]

Reviewers' comments:

Reviewer #1 (Remarks to the Author):

In this study, Zhou et al. analyzed the transcriptomes from the initial apical cell and basal cell lineages of early Arabidopsis embryos. I found the work to be technically well done and the authors provided a good description of the transcriptome dynamics during these lineage specification events.

However, my major criticism is that while this detailed description was good, the claims made in the title and abstract are not fully substantiated by the results in the manuscript. I do not think describing genome-wide transcript dynamics reveals the molecular basis for cell fate specification. Rather, it mostly catalogs the response to these initial specification events. Similarly, the identification of lncRNAs and alternative splicing present in embryos does not reveal how they influence lineage specification as was claimed in manuscript. In general, I think the report would be improved if the authors focused on at least one of several possible differentiation mechanisms (e.g. lncRNAs, alternative splicing and TFs) and performed functional studies to reveal mechanistic basis.

Another major criticism is that I thought the authors should have written more describing previous attempts to examine apical cell and basal cell transcriptomes, and comparing their own datasets with these. Such studies have been published by three independent groups over the last six years. For example, Belmonte et al. (2013, PNAS) performed microarrays on laser capture microdissected embryo proper and suspensor at the globular/32-cell stage; Slane et al. (2014, Development) used fluorescent nuclei sorting on apical and basal cell lineages between the 1-cell and 32-cell stage; Palovaara et al. (Nat. Plants, 2017) used INTACT to isolate RNA from ACL and BCL for microarrays. I think these studies should be described more in the introduction and discussion to put the work of Zhou et al. in better context. Importantly various cross-comparisons between the presented work and this previously published work should also be done to make it clear what is novel in this manuscript.

A final major criticism is that the independent verification of their dataset was not sufficient in my opinion. It was not clear to me how many ACL and BCL markers that they tested with gene fusions to GFP. Were only two candidates each tested? Or were more than two tested and these were then only two that had detectable GFP? Because the main result of this manuscript is the ACL and BCL transcriptomes, I think more validation needs to be performed for independent verification.

Minor comments

1. Supp. Fig. 2: The authors may want to comment on why the BC 1, 2 and 3 samples are equally enriched in embryo proper and suspensor markers. This seems unexpected, but interesting.
2. Lines 144-146: I don't think the small portion of upregulated genes in ACL and BCL that overlap suggests that the zygotic genome is activated during ACL/BCL specification. The zygotic genome is activated in the zygote and the observed differences likely just reflect lineage-specific differentiation.
3. Line 220: 717 novel lncRNAs (out of 990 total) is a high number in my opinion. Is it possible for the authors to verify a few of these independently or cross-check their bioinformatics pipelines to determine how many false positives are among the novel lncRNAs?
4. Lines 220-222: It was not clear to me what was meant by the lncRNAs being "...primarily generated from the sense and antisense strands...". Do the authors mean the sense and antisense strands of protein-coding genes?
5. Lines 223-225: I disagree with the assumption that because lncRNAs can be grouped similarly as protein-coding genes then they are implied to be involved in lineage specification. Functional studies would need to be done to make this point.
6. Lines 241-244: I also disagree that the identification of lineage-specific alternative splicing

indicates that they are critical for lineage specification during embryogenesis. Functional studies would need to be done to make this point.

7. Lines 269-270: I am not sure how meaningful this sentence (and above two paragraphs) are for this study. It is well accepted that TFs contribute to ACL and BCL specification, and this has been especially demonstrated for the BCL as the authors wrote.

8. Lines 345-347: Could the reason that BCL is more similar to zygotes be because a larger proportion of the BCL is derived from the asymmetrically dividing zygote compared to ACL?

9. I didn't understand the main points of the final two paragraphs of the Discussion. Maybe these can be written more clearly and concisely to get the intended points across?

Reviewer #2 (Remarks to the Author):

In their manuscript, Zhou and colleagues studied early embryo transcriptomes in Arabidopsis. They isolated the apical and basal cells of 2-cell stage embryos and the embryo proper and suspensor of 32-cell stage embryo, and analyzed their transcriptomes. They compared these transcriptomes with their previously published transcriptomes of egg cell and zygote. Comparisons showed that apical and basal cells have distinct transcriptomes, presumably due to both activation and degradation of specific sets of transcripts. Separation of apical and basal cells at 2-cell stage is technically very challenging and this research provides a very valuable resource. However, the resource will need further experimental validation to be useful to the plant embryogenesis research community. The present study unfortunately does not go beyond describing the transcriptomes and speculations on how the apical cell and basal cell acquire their identities. Experimental support for these statements or for the molecular regulation that are proposed are lacking. Furthermore, there are some concerns with interpretation of the data, and the writing is less than clear. While the work is interesting, at this stage it appears too preliminary for publication.

Detailed comments:

1. Title: "molecular base", maybe "the molecular basis" is more suitable here.

2. Line 20: "may involve in" should be "may be involved in"

3. Line 22: "molecular base" should be "the molecular basis"

4. Line 27-33: "In zygotes of most angiosperm, ...". The information is incorrect and misleading. In many angiosperms, the basal cell is the same size or smaller than the apical cell (e.g. "Embryology of Angiosperms", B.M. Johri, 1984).

5. Line 60-62: "Here we apply this technique to generate a transcriptional map of the ACL and BCL of early proembryos from the 1- to 32-cell embryo stage."

Here, the statement is easily misinterpreted. Only 1-cell and 32-cell stages were used, not the range of stages.

6. Line 73: "The proembryos was" should be "The proembryos were"

7. Line 79: It is not easy to find the correlation coefficients between different replicates among the same cell type from Fig. S1, maybe the authors can use four red frames to highlight this for better visualization for readers.

8. Line 113-116: Regarding the relationship between the zygote and its two daughter cells, BC displayed a closer relationship to the zygote than to AC (Fig. 1b,d), suggesting that BC and zygote have similar intrinsic gene expression profiles. This is hard to derive from the tree and correlation

plots. In fact, this is not at all very convincing.

9. Line 133-134: Genes that are down-regulated in 32E compared with AC are not mentioned here.

10. Line 135-144: It is true that we can explain the huge differences between BC and 32S based on the many more up-regulated genes (5,124) compared with other comparisons. However, the same method can't be used to explain that BC is much more similar to the zygote (up: 2,791; down: 3,764) than AC (up: 3,002; down: 3,166). Is it possible to also consider the extent of gene expression change other than by counting the number of differentially expressed genes?

11. Line 174-175: "Also, the differences in their expression between the ACL and BCL increased as development progressed.". Evidence (figures or tables) is missing to support this statement.

12. Line 175-192: Based on the method that is used to identify the ACL- and BCL-maintained genes, there is no doubt that ACL- and BCL-maintained genes have a low expression in the BCL and ACL, respectively. So, it is more clear to cluster the ACL- and BCL-maintained gene expression across EC-Zy14-Zy24-AC-32E and EC-Zy14-Zy24-BC-32S, respectively.

13. Line 186-188: "also clustered into" should be "were clustered into"

14. Line 175-192: There are 370 and 209 genes that are specifically de novo transcribed in the ACL and BCL, respectively. And, there are 539 and 723 genes that are specifically downregulated in the BCL and ACL, respectively. These results showed that there were less genes being de novo transcribed and specifically downregulated in the BCL compared with the ACL. Can this explain that BC is more similar to zygotes than AC at the transcriptional level?

15. Line 193-201: In Fig. 4 and 5, to validate the transcriptome analysis results, reporters were generated and analyzed. The authors should provide full information about these experiments. How many genes were tested and what was the percentage of patterns that confirmed their transcriptome results?

16. Line 217: "of mean length 1183 nt" should be "with a mean length of 1,183 nt"

17. Line 212-240: The jump to lncRNA and alternative splicing is very abrupt. What are the percentages of transcripts found in other developmental pathways? Are these two specifically needed for embryogenesis in other organisms? More information is needed to integrate this results in the text. Ideally, there would be genetic data to support the importance of lncRNA's and/or alternative splicing in embryo development, particularly in specifying apical and basal lineages.

18. Fig. 6a: It might be better to use the same color or darker of curve for corresponding cell type name instead of black.

19. Fig. 7i: It might be better to use the blue color as in f and g for BC instead of yellow.

20. Line 301-303: It is interesting to identify how many and which TFs are inherited from zygotes in both ACL and BCL. Are the binding motifs of these inherited TFs enriched in the promoters of transcripts that derive from cell type-specific de novo transcription. Is there a possibility that the uneven inheritance of transcripts contributes to the cell type-specific de novo transcription?

21. Line 326-327: "The regulatory mechanism of lineage specification is more complex than formerly believed.". Not sure this is true... we may have always considered this to be a complex process, just lacking actual evidences both in experimental and bioinformatics studies.

22. Line 348-360: Results of EMB genes expression should go to result part, instead of discussion.

23. Line 364-372: It is unclear what the two models are and what kind of questions they would like to explain using these two models.

Reviewer #3 (Remarks to the Author):

The ms by Zhao et al. addresses a fundamental problem of plant developmental biology – the establishment of embryo fate during sexual reproduction in flowering plants, using *Arabidopsis thaliana* as a model. Following asymmetric cell division of the zygote, the apical daughter takes on embryo fate whereas the basal daughter essentially gives rise to an extraembryonic suspensor that anchors the developing embryo within the ovule. There have been previous attempts to define the two different cell lineages by expression profiling. However, what sets this study apart is the scope (an order of magnitude more genes analysed) and the manual separation of cells in early embryogenesis. This enabled expression profiling of apical vs. basal daughter cell as well as comparison of apical (ACL) and basal (BCL) lineage at the 32-cell stage of embryogenesis. RNAseq generated expression profiles and led to the identification of differentially expressed genes. These were grouped into different classes depending on their developmental, i.e. temporal profiles, and validated by GFP fusion expression analysis in a few cases. This is all well documented with the *Arabidopsis* gene identifiers indicated except for the validated genes shown; ACL1 & ACL2 and BCL1 & BCL2 – here the AT gene identifiers should be added). Another minor criticism: it is not clear from the data how representative the examples shown really are.

The authors also performed a GO terms analysis that revealed differences between A and B lineages (e.g. cell division in A but cell death in B). In addition, gene regulatory networks were inferred from differentially expressed transcription factor (TF) genes and matching putative target genes (identified by putative TF-binding sites in their promoters).

The main conclusions supported by the expression analysis are:

- The B cell is similar to the zygote whereas the A cell is different. This is shown here for the first time and consistent with the experimentally induced change from B to A but not vice versa as reported previously.
- The A lineage is established early ('embryo fate') whereas later changes in the B lineage occur towards differentiation of the extraembryonic suspensor.

In addition to these highly relevant findings, the manuscript also provides a rich resource for further studies in early embryogenesis.

I strongly recommend publication of this manuscript.

Response to Reviewer #1

Q1. In this study, Zhou et al. analyzed the transcriptomes from the initial apical cell and basal cell lineages of early Arabidopsis embryos. I found the work to be technically well done and the authors provided a good description of the transcriptome dynamics during these lineage specification events. However, my major criticism is that while this detailed description was good, the claims made in the title and abstract are not fully substantiated by the results in the manuscript. I do not think describing genome-wide transcript dynamics reveals the molecular basis for cell fate specification. Rather, it mostly catalogs the response to these initial specification events. Similarly, the identification of lncRNAs and alternative splicing present in embryos does not reveal how they influence lineage specification as was claimed in manuscript. In general, I think the report would be improved if the authors focused on at least one of several possible differentiation mechanisms (e.g. lncRNAs, alternative splicing and TFs) and performed functional studies to reveal mechanistic basis.

Reply: Thanks a lot for your interest in our work. Apical and basal cell lineage specification is an important topic in the field of plant embryogenesis. However, genome-wide transcriptome analysis of initial stages of apical cell and basal cell lineages of early Arabidopsis embryos has not been achieved due to technical limitation. In the present study, by overcoming technical limitations, we aimed to answer this question by genome-wide transcriptome analysis. From transcriptome analysis, we found several important clues for exploring the mechanism of lineage specification, which will provide valuable source data to promote the research in this field. According to your suggestion, the title and abstract have been revised to make our conclusion more accurate. “molecular basis” have been removed in both title and abstract. Since molecular mechanism regulating apical and basal cell lineage specification are largely unknown, analysis of lncRNAs and alternative splicing in the present study aimed to investigate potential mechanism for apical and basal cell lineage specification from different angles. From our detailed transcriptome analysis, we found several possible mechanisms including TFs, lncRNAs and alternative splicing for lineage specification, which are worthy to be investigated in detail through functional studies in the future researches. It will be our next project. At present, we try to publish this important transcriptome analysis results, which provide valuable clues and thus help researches from different groups in this field to investigate the molecular mechanism for differentiation together. Hence, to express the conclusion more accurately, the claims about lncRNAs and alternative splicing in lineage specification in the manuscript have also been revised in the manuscript.

Q2. Another major criticism is that I thought the authors should have written more describing

previous attempts to examine apical cell and basal cell transcriptomes, and comparing their own datasets with these. Such studies have been published by three independent groups over the last six years. For example, Belmonte et al. (2013, PNAS) performed microarrays on laser capture microdissected embryo proper and suspensor at the globular/32-cell stage; Slane et al. (2014, Development) used fluorescent nuclei sorting on apical and basal cell lineages between the 1-cell and 32-cell stage; Palovaara et al. (Nat. Plants, 2017) used INTACT to isolate RNA from ACL and BCL for microarrays. I think these studies should be described more in the introduction and discussion to put the work of Zhou et al. in better context. Importantly various cross-comparisons between the presented work and this previously published work should also be done to make it clear what is novel in this manuscript.

Reply: Thank you a lot for suggestion. More descriptions about previous results from three relevant papers have been added in both introduction and discussion part (Line 54-62; 313-330).

Cross-comparisons between our present data and previously published work have also been performed (Supplementary Fig. 9). By the comparisons, the novelties of our present work were enhanced. First, our present manuscript is the first study for the transcriptome analysis of apical cell and basal cell in Arabidopsis. Second, our present manuscript investigated the cellular transcriptome of embryo proper and suspensor, which could provide expression information from different angle when compared with the microarray data of nuclei from embryo proper or suspensor. Third, previous cellular and nuclear transcriptome data were generated by microarray technology, whereas RNA-seq technology was used to develop transcriptome data in the present study. RNA-seq has advantages of high sensitivity and reproducibility, resulting in that more expressed genes were identified in the transcriptomes. Another significant advantage is that RNA-seq could be used to identify alternative isoforms and previously unannotated lncRNAs as shown in the manuscript. Fourth, we performed hand emasculum and pollination to isolate embryo at 1-cell and 32-cell stages. Early embryos at corresponding stages were isolated and collected one by one under inverted microscope, which ensure our transcriptome data are representative of specific embryo stages and free of contamination from seed coats and endosperms. Importantly this work fill in the gap of transcriptome dynamic during apical and basal cell lineage specification, which is required for understanding molecular mechanism underlying cell lineage specification. According to your suggestion previous works have been briefly described in both introduction and discussion sections and the comparison of the data with previous data is shown in new figure (Supplementary Fig. 9).

Q3: A final major criticism is that the independent verification of their dataset was not sufficient in my opinion. It was not clear to me how many ACL and BCL markers that they tested with gene

fusions to GFP. Were only two candidates each tested? Or were more than two tested and these were then only two that had detectable GFP? Because the main result of this manuscript is the ACL and BCL transcriptomes, I think more validation needs to be performed for independent verification.

Reply: The descriptions about ACL and BCL markers selected for verification have been added in the revised manuscript. In fact, we choose twenty differentially expressed genes between ACL and BCL for GFP reporter verification (Ten candidates for each cell lineage). The results revealed that the expression patterns of selected candidates are highly consistent with the transcriptome analysis results. So we selected four of them displayed in our manuscript. According to your suggestion, the results of another sixteen marker lines for ACL and BCL have been added (Supplementary Fig. 5-7), and relevant descriptions have also been added in the revised our manuscript.

Q4. Supp. Fig. 2: The authors may want to comment on why the BC 1, 2 and 3 samples are equally enriched in embryo proper and suspensor markers. This seems unexpected, but interesting.

Reply: Yes, this is unexpected and very interesting. This may indicate that BC has a relatively similar intrinsic gene expression profile with zygote, whereas the AC showed distinct enrichment in embryo proper and suspensor markers, suggesting that BC is also inherited genetic information for embryogenesis from the zygote and is not really switched to embryo or suspensor pathway yet, but the AC already showed something special for embryogenesis. In fact, consistent with this result, early reports demonstrated that basal cell still has embryogenesis potential, which could divide to form a typical embryo after destroying the apical cell (Gooh, K., et al. 2015). Based on these new findings we suggest that there might be the possibility of embryogenesis initiation in apical cell, but not in zygote. However, we can only propose the idea and we can't clearly explain the phenomenon yet due to limited evidences. That's why we do not discuss the issue in the manuscript. Further investigation in near future should allow to demonstrate the proposal.

Q5. Lines 144-146: I don't think the small portion of upregulated genes in ACL and BCL that overlap suggests that the zygotic genome is activated during ACL/BCL specification. The zygotic genome is activated in the zygote and the observed differences likely just reflect lineage-specific differentiation.

Reply: To express it more accurately, the sentence "developmental stage-dependent activation of the zygotic genome during ACL and BCL specification" has been revised as "developmental stage-dependent transcription during ACL and BCL specification" in the manuscript (Line 148-150).

Q6. Line 220: 717 novel lncRNAs (out of 990 total) is a high number in my opinion. Is it possible for the authors to verify a few of these independently or cross-check their bioinformatics pipelines to determine how many false positives are among the novel lncRNAs?

Reply: To detect lncRNAs in early embryos, we applied five independent programs: CNCI (v2), CPC (v0.9-r2), PLEK (v1.2), CPAT (v1.2.2) and pfam_scan (v1.6-1) to determine lncRNAs from assembled transcripts. Transcripts that passed through at least four independent programs were annotated as lncRNAs. Then, 717 novel lncRNAs were identified in early embryos. Considering that 6510 *Arabidopsis* lncRNAs were annotated in the *Arabidopsis* seedlings (Zhao et al., 2018) and lncRNAs in early proembryos have not yet fully investigated, 717 novel lncRNAs in early embryos seems not a high number.

To further confirm these annotated lncRNAs, the sequence of twenty lncRNA candidates were selected for verification by RT-PCR and Sanger sequencing. All the transcripts of predicated lncRNAs could be successfully amplified by RT-PCR (Supplementary Fig. 8). Confirmed lncRNA sequences are highly consistent with the assembled lncRNA sequences (Supplementary Data 6), confirming the high reliability of predicated lncRNAs from our cell lineage-specific transcriptome. These results have been added in the supplementary material.

Q7. Lines 220-222: It was not clear to me what was meant by the lncRNAs being “...primarily generated from the sense and antisense strands...”. Do the authors mean the sense and antisense strands of protein-coding genes?

Reply: Yes, you are right. Sense lncRNAs are transcribed from the sense strand of protein-coding genes. The antisense lncRNAs are transcribed from the antisense strand of protein-coding genes. Related information has been added in the revised manuscript (Line 244-247).

Q8. Lines 223-225: I disagree with the assumption that because lncRNAs can be grouped similarly as protein-coding genes then they are implied to be involved in lineage specification. Functional studies would need to be done to make this point.

Reply: The transcriptomes of apical and basal cell lineages of early proembryos could be grouped similarly as protein-coding genes based on the expression of lncRNAs. In addition, a series of lineage-specific lncRNAs have also been identified, suggesting potential roles of these lineage-specific lncRNAs in embryo differentiation. Although detailed roles of lineage-specific lncRNAs in early embryogenesis remain unknown, it is worthy to be investigated in the further studies. Here, we aimed to provide possible clues for exploring the mechanism of lineage specification from different aspects, which will promote the researches in this field to investigate the molecular mechanism regulating lineage specification. To make conclusion more accurately,

the descriptions about the roles of lncRNAs in lineage specification have been revised in our manuscript.

Q9. Lines 241-244: I also disagree that the identification of lineage-specific alternative splicing indicates that they are critical for lineage specification during embryogenesis. Functional studies would need to be done to make this point.

Reply: Similar to the point 8, to express more accurately, we have revised the expression about the roles of lineage-specific alternative splicing isoforms in lineage specification during embryogenesis.

Q10. Lines 269-270: I am not sure how meaningful this sentence (and above two paragraphs) are for this study. It is well accepted that TFs contribute to ACL and BCL specification, and this has been especially demonstrated for the BCL as the authors wrote.

Reply: As you pointed out, a few TFs have been shown to be involved in ACL and BCL development, especially on basal cell lineage development. However, the roles of most other TFs in ACL and BCL specification, especially about their contribution to different transcription program in apical and basal cell lineages and apical and basal cell differentiation shortly after zygote division are still largely unknown. It is worthy to pay more attention to these lineage-specific TFs in the further studies, which will lead to a better understanding of the fundamental mechanisms of ACL and BCL specification. According to your comments we have modified this sentence (Line 304-306).

Q11. Lines 345-347: Could the reason that BCL is more similar to zygotes be because a larger proportion of the BCL is derived from the asymmetrically dividing zygote compared to ACL?

Reply: Yes, it's a possible reason. However, we do not have enough evidences to support this claim. We hope the fact itself will help to analyze the apical and basal cell fate specification.

Q12. I didn't understand the main points of the final two paragraphs of the Discussion. Maybe these can written more clearly and concisely to get the intended points across?

Reply: Analysis of the expression pattern of *EMB* genes in the process of lineage specification aimed to test whether most genes required for embryo development are preferentially expressed in the apical cell lineage. According to the comments from reviewer#2, the results about the expression pattern of *EMB* genes have been moved into the result section (Line 165-176). Relevant explanation has also been added in our revised manuscript.

The last paragraph of the discussion aimed to discuss the initiation stage of plant embryogenesis.

This paragraph has been revised to make the main point clearer.

Response to Reviewer #2

In their manuscript, Zhou and colleagues studied early embryo transcriptomes in Arabidopsis. They isolated the apical and basal cells of 2-cell stage embryos and the embryo proper and suspensor of 32-cell stage embryo, and analyzed their transcriptomes. They compared these transcriptomes with their previously published transcriptomes of egg cell and zygote. Comparisons showed that apical and basal cells have distinct transcriptomes, presumably due to both activation and degradation of specific sets of transcripts. Separation of apical and basal cells at 2-cell stage is technically very challenging and this research provides a very valuable resource. However, the resource will need further experimental validation to be useful to the plant embryogenesis research community. The present study unfortunately does not go beyond describing the transcriptomes and speculations on how the apical cell and basal cell acquire their identities. Experimental support for these statements or for the molecular regulation that are proposed are lacking. Furthermore, there are some concerns with interpretation of the data, and the writing is less than clear. While the work is interesting, at this stage it appears too preliminary for publication.

Reply: Thank a lot for your interest in our work. Your comments have greatly helped us to improve our manuscript. Apical and basal cell lineage specification is an important development event in early embryogenesis. Although great efforts have been made to investigate the transcriptome of apical and basal cell lineages of early proembryos, the molecular mechanism regulating lineage specification are still largely unknown, mainly because technical difficulty in obtaining apical cell and basal cell and little gene expression data are available. Here, by overcoming technical limitations, we successfully separate apical and basal cells for transcriptome analysis, which will provide very valuable resources for scientists in the field of plant development. According to your suggestion, the reliability of transcriptome data were further confirmed by GFP fusion reporter line analysis, RT-PCR and Sanger sequencing. These data have been added in our revised manuscript (Supplementary Fig. 5-8 and Supplementary Data 6). Functional analysis of these lineage-specific TFs, lncRNA and isoforms is really a big project, which will take years to a clear conclusion although only few genes can be investigated. In the revised manuscript we try to concentrate more on the transcriptome analysis results.

Q2. Title: “molecular base”, maybe “the molecular basis” is more suitable here.

Reply: According to the comments of reviewer#1, the title has been revised as “Cell lineage-specific transcriptome analysis for interpreting cell fate specification of proembryos”. “molecular base” is not used in the current title.

Q3. Line 20: “may involve in” should be “may be involved in”

Reply: “may involve in” has been revised as “may be involved in” in the revised manuscript (Line 20).

Q4. Line 22: “molecular base” should be “the molecular basis”

Reply: “molecular base” has been revised as “the molecular pathways” in the revised manuscript according to the comments from Reviewer#1 (Line 22).

Q5. Line 27-33: “In zygotes of most angiosperm,”. The information is incorrect and misleading. In many angiosperms, the basal cell is the same size or smaller than the apical cell (e.g. “Embryology of Angiosperms”, B.M. Johri, 1984).

Reply: To express more accurately, this sentence has been revised according to your comments. The word “most” has been deleted in the revised manuscript. The species name has also been added in the revised sentences (Line 27-33).

Q6. Line 60-62: “Here we apply this technique to generate a transcriptional map of the ACL and BCL of early proembryos from the 1- to 32-cell embryo stage.” Here, the statement is easily misinterpreted. Only 1-cell and 32-cell stages were used, not the range of stages.

Reply: “.....early proembryos from the 1- to 32-cell embryo stage.” has been revised as “..... early proembryos at 1-cell and 32-cell embryo stage.” (Line 64-66).

Q7. Line 73: “The proembryos was” should be “The proembryos were”

Reply: “The proembryos was” has been revised as “The proembryos were” (Line 78).

Q8. Line 79: It is not easy to find the correlation coefficients between different replicates among the same cell type from Fig. S1, maybe the authors can use four red frames to highlight this for better visualization for readers.

Reply: According to your suggestion, four magenta frames have been used to highlight different replicates of the same cell type in the revised Fig. S1.

Q9. Line 113-116: Regarding the relationship between the zygote and its two daughter cells, BC displayed a closer relationship to the zygote than to AC (Fig. 1b, d), suggesting that BC and zygote have similar intrinsic gene expression profiles. This is hard to derive from the tree and correlation plots. In fact, this is not at all very convincing.

Reply: To make the results more convincing, the exact Pearson correlation coefficients between

different samples have been calculated and added in the revised Figure 1d. From the Pearson correlation coefficients, the transcriptome of basal cell is relatively more similar to zygote than the apical cell.

Q10. Line 133-134: Genes that are down-regulated in 32E compared with AC are not mentioned here.

Reply: The information of downregulated genes in 32E compared with AC has been added in the revised manuscript (Line 137-139).

Q11. Line 135-144: It is true that we can explain the huge differences between BC and 32S based on the many more up-regulated genes (5,124) compared with other comparisons. However, the same method can't be used to explain that BC is much more similar to the zygote (up: 2,791; down: 3,764) than AC (up: 3,002; down: 3,166). Is it possible to also consider the extent of gene expression change other than by counting the number of differentially expressed genes?

Reply: Thanks for your suggestion. The extent of gene expression change has also been considered in our revised manuscript. Differentially expressed genes were clustered into different groups according to the fold change, and relevant data have been added in the revised Figure 2 (c, d). This method could also support the huge differences between BC and 32S compared with other comparisons. However, comparable number of differentially expressed genes in different groups were found in the two comparisons (AC/Zy24 and BC/Zy24), which could not be used to explain that BC is much more similar to the zygote. Hence, we use Pearson correlation coefficient and unsupervised hierarchical clustering, which are two common methods for determining the relationships between different samples, to analyze the relationship between the zygote and its two daughter cells in the manuscript. Pearson correlation coefficients between different samples have been calculated and added in the revised Figure 1d. Both Pearson correlation coefficients and unsupervised hierarchical clustering result support that the transcriptome of basal cell is more similar to zygote than the apical cell.

Q12. Line 174-175: "Also, the differences in their expression between the ACL and BCL increased as development progressed.". Evidence (figures or tables) is missing to support this statement.

Reply: According to your suggestion, the figure that support the statement has been added in the revised Supplementary Figure 4a. As shown in the Figure, the number of differentially expressed genes between 32E and 32S is much larger than that between AC and BC.

Q13. Line 175-192: Based on the method that is used to identify the ACL- and BCL-maintained genes, there is no doubt that ACL- and BCL-maintained genes have a low expression in the BCL and ACL, respectively. So, it is more clear to cluster the ACL- and BCL-maintained gene expression across EC-Zy14-Zy24-AC-32E and EC-Zy14-Zy24-BC-32S, respectively.

Reply: Thanks for your suggestion. We have added the results of expression profile of the ACL- and BCL-maintained genes across EC-Zy14-Zy24-AC-32E and EC-Zy14-Zy24-BC-32S in the supplementary material according to your suggestion (Supplementary Figure 4b, c).

From our results, both *de novo* transcription in the apical cell lineage and the downregulation of certain genes in the basal cell lineage contribute to the expression of ACL- maintained genes. Similarly, both *de novo* transcription in the basal cell lineage and downregulation of certain genes in the apical cell lineage contribute to the expression of BCL- maintained genes. The reason for specific expression pattern of lineage-maintained genes could be easily discovered from the cluster results across EC-Zy14-Zy24-AC-BC-32E-32S. So we still keep the results in the manuscript for reader's reference.

Q14. Line 186-188: “also clustered into” should be “were clustered into”

Reply: “also clustered into” has been revised as “were clustered into” (Line 205).

Q15. Line 175-192: There are 370 and 209 genes that are specifically *de novo* transcribed in the ACL and BCL, respectively. And, there are 539 and 723 genes that are specifically downregulated in the BCL and ACL, respectively. These results showed that there were less genes being *de novo* transcribed and specifically downregulated in the BCL compared with the ACL. Can this explain that BC is more similar to zygotes than AC at the transcriptional level?

Reply: We actually agree with you. However, since this section are focused on the expression pattern of cell lineage-maintained genes, other differentially expressed genes between apical and basal cell are not analyzed here.

Q16. Line 193-201: In Fig. 4 and 5, to validate the transcriptome analysis results, reporters were generated and analyzed. The authors should provide full information about these experiments. How many genes were tested and what was the percentage of patterns that confirmed their transcriptome results?

Reply: Detailed information for GFP fusion protein analysis have been added in the revised manuscript. Gene ID and primer sequences for each candidate have been listed in the supplementary Table 2. And the description about GFP fusion reporter for validation has also added in the revised manuscript. We choose twenty marker genes for GFP fusion reporter

verification (Ten candidates for each cell lineage). The results revealed that the expression patterns of selected candidates are highly consistent with the transcriptome analysis results. So we selected four of them displayed in our manuscript. According to the comments from you and reviewer #1, the results of another sixteen marker lines for ACL and BCL have been added in the revised our manuscript (Supplementary Figure 5-7).

Q17. Line 217: “of mean length 1183 nt” should be “with a mean length of 1,183 nt”

Reply: “of mean length 1183 nt” has been revised as “with a mean length of 1,183 nt” (Line 244-245).

Q18. Line 212-240: The jump to lncRNA and alternative splicing is very abrupt. What are the percentages of transcripts found in other developmental pathways? Are these two specifically needed for embryogenesis in other organisms? More information is needed to integrate this results in the text. Ideally, there would be genetic data to support the importance of lncRNA's and/or alternative splicing in embryo development, particularly in specifying apical and basal lineages.

Reply: In animals, lncRNAs and alternative splicing in embryos have been well characterized, suggesting potential roles of them in embryo development. Functional analysis revealed that both lncRNAs and alternative splicing play important roles in multiple developmental processes such as maintaining embryonic stem cell identity (Lu et al., 2014; Luo et al., 2016; Pauli et al., 2011). In plants, lncRNAs and alternative splicing have also been studied in the normal developmental and stress conditions. And important roles of lncRNA in different developmental processes such as photomorphogenesis, flowering time control and male fertility have been confirmed through functional studies (Ding et al., 2012; Heo and Sung, 2011; Wang et al., 2014; Zhou et al., 2012). However, the expression and function of lncRNAs and alternative splicing in early embryos remains unknown. So we analyzed lncRNA and alternative splicing in apical and basal cell lineages of early proembryos to provide valuable resources for exploring the mechanism of cell lineage differentiation. Functional analysis of these lineage-specific lncRNAs and isoforms is really a big project, which will take years to get a clear conclusion. It will be our next project. According to your comments, several relevant descriptions have been added in our revised manuscript (Line 234-244; 263-270).

Q19. Fig. 6a: It might be better to use the same color or darker of curve for corresponding cell type name instead of black.

Reply: According to your suggestion, same color has been used for the cell type name.

Q20. Fig. 7i: It might be better to use the blue color as in f and g for BC instead of yellow.

Reply: The blue color has been used for Figure 7i according to your suggestion.

Q21. Line 301-303: It is interesting to identify how many and which TFs are inherited from zygotes in both ACL and BCL. Are the binding motifs of these inherited TFs enriched in the promoters of transcripts that derive from cell type-specific *de novo* transcription. Is there a possibility that the uneven inheritance of transcripts contributes to the cell type-specific *de novo* transcription?

Reply: According to your suggestion, TFs that are likely inherited from zygote have been identified in apical and basal cell. Binding motifs of these inherited TFs were identified in the promoters of cell type-specific *de novo* genes. Although the binding motifs of these inherited TFs are not enriched through significance test, there is still a possibility that these inherited TFs may play a role in triggering a cascade of events leading to activating cell-type specific transcription program. It will be very interesting to work on these uneven inherited transcripts that may contribute to the cell type-specific *de novo* transcription.

Q22. Line 326-327: “The regulatory mechanism of lineage specification is more complex than formerly believed.”. Not sure this is true... we may have always considered this to be a complex process, just lacking actual evidences both in experimental and bioinformatics studies.

Reply: “The regulatory mechanism of lineage specification is more complex than formerly believed.” has been removed in the revised manuscript according to your comments.

Q23. Line 348-360: Results of *EMB* genes expression should go to result part, instead of discussion.

Reply: The results of *EMB* genes expression have been moved into the result section according to your suggestion (Line 165-176).

Q24. Line 364-372: It is unclear what the two models are and what kind of questions they would like to explain using these two models.

Reply: In this section, we aimed to discuss two possibilities of embryogenesis initiation. Currently, it seems accepted that embryogenesis is initiated in zygote and zygote division is the starting point of the embryogenesis. However, there is no solid evidence for this conclusion. According to our transcriptome analysis in this work, it seems different from current opinion. This is a critical question since in different cases our strategy for the investigation of molecular mechanism triggering embryogenesis will be modified accordingly.

In fact, there is a major difference of early embryogenesis between animal and plant. In animals,

both daughter cells of zygote will contribute to a mature embryo, whereas, in higher plants two daughter cells of zygote show differential contributions to the embryonic development. The smaller apical cell divide to form the major parts of a mature embryo, whereas the larger basal cell usually undergoes limited divisions to form a suspensor composed of a few cells and only the uppermost suspensor cell will differentiate into an embryo hypophysis, and other suspensor cells will degenerate via programmed cell death. Based on our transcriptome data and our analysis we proposed two possibilities of embryogenesis initiation: the first is that plant embryogenesis initiates at zygotic stages, similar to that of animals; and the second is that plant embryogenesis initiates in apical cell at 2-cell proembryo stage, different from that of animals. We hope these two models could promote further discussion about molecular mechanism triggering embryogenesis or help us to consider that from which cell or developmental stage we could find those switch factors for guiding the entry of embryogenesis pathway.

Response to Reviewer #3

Q1. This is all well documented with the Arabidopsis gene identifiers indicated except for the validated genes shown; ACL1 & ACL2 and BCL1 & BCL2 – here the AT gene identifiers should be added).

Reply: Thank a lot for your interest on our work. Gene ID and primer sequences for each candidate have been listed in the supplementary Table 2.

Q2. Another minor criticism: it is not clear from the data how representative the examples shown really are.

Reply: Descriptions about representative examples for validation have added in the revised manuscript. We choose twenty examples for GFP fusion reporter verification (Ten candidates for each cell lineage). The results revealed that the expression patterns of selected candidates are consistent with the transcriptome analysis results. So we selected four of them displayed in our manuscript. According to the comments from you and reviewer #1, the results of another sixteen marker lines for ACL and BCL have been added in the revised our manuscript (Supplementary Fig. 5-7).

References:

- Belmonte, M.F., *et al.* (2013). Comprehensive developmental profiles of gene activity in regions and subregions of the Arabidopsis seed. *Proc. Natl. Acad. Sci. USA* *110*, E435-444.
- Ding, J., *et al.* (2012). A long noncoding RNA regulates photoperiod-sensitive male sterility, an essential component of hybrid rice. *Proc. Natl. Acad. Sci. USA* *109*, 2654-2659.
- Heo, J.B., and Sung, S. (2011). Vernalization-mediated epigenetic silencing by a long intronic noncoding RNA. *Science* *331*, 76-79.
- Lu, X., *et al.* (2014). The retrovirus HERVH is a long noncoding RNA required for human embryonic stem cell identity. *Nat. Struct. Mol. Biol.* *21*, 423-425.
- Luo, S., *et al.* (2016). Divergent lncRNAs Regulate Gene Expression and Lineage Differentiation in Pluripotent Cells. *Cell Stem Cell* *18*, 637-652.
- Gooh, K., *et al.* (2015). Live-cell imaging and optical manipulation of Arabidopsis early embryogenesis. *Dev. Cell* *34*, 242-251.
- Pauli, A., *et al.* (2011). Non-coding RNAs as regulators of embryogenesis. *Nat. Rev. Genet.* *12*, 136-149.
- Schon, M.A., and Nodine, M.D. (2017). Widespread Contamination of Arabidopsis Embryo and Endosperm Transcriptome Data Sets. *Plant Cell* *29*, 608-617.
- Wang, Y., *et al.* (2014). Arabidopsis noncoding RNA mediates control of photomorphogenesis by red light. *Proc. Natl. Acad. Sci. USA* *111*, 10359-10364.
- Zhao, X., *et al.* (2018). Global identification of Arabidopsis lncRNAs reveals the regulation of MAF4 by a natural antisense RNA. *Nat. Commun.* *9*, 5056.
- Zhou, H., *et al.* (2012). Photoperiod- and thermo-sensitive genic male sterility in rice are caused by a point mutation in a novel noncoding RNA that produces a small RNA. *Cell Res.* *22*, 649-660.

REVIEWERS' COMMENTS:

Reviewer #1 (Remarks to the Author):

In their revised manuscript, Zhou et al. have corrected their title/abstract, included more background on previous attempts to profile apical cell and basal cell transcriptomes, and have also provided additional data and controls to confirm that their resource is of high-quality. The writing is much improved, and as I mentioned in my original review, this appears to be a high-quality dataset that should help elucidate the molecular pathways for these initial cell lineages. And although I appreciate the authors addressing most of my points, my major criticism (#1) was that describing the different amounts of lncRNAs, alternative splicing and transcript levels in apical and basal cell lineages does not address how these cell lineages become different. I realize that such functional studies would require more experiments, but this major point was not addressed and I expect a study to have functional data to be published in a journal like Nature Communications.

Reviewer #2 (Remarks to the Author):

I appreciate the efforts the authors made in response to my questions and minor concerns. Specially, they provided information on additional markers analysis to strengthen their data validation. However, given a decent number of prior examples, it is not at all surprising that the apical and basal cells have distinct transcriptomes. While I applaud the effort that went into this work and the resource it provides to a specialized community, it remains a description of transcriptomes with some technical validation, without any hint of its genetic regulation or functional implications. Furthermore, there still is a lack of substantial evidence to support the speculations about the potential functions of TF, lncRNA, and alternative splicing in cell fate specification during plant embryogenesis, which should have been the highlight of the manuscript. Even though lncRNA and alternative splicing are very important for animal embryogenesis as explained by authors, it does not necessarily mean that they will have important functions in cell specification during plant embryogenesis. Based on the above, I still regard this too preliminary to be published as a standalone paper, rather than a resource.

Minor point:

The information about WOX8 expression in line 353 does not appear to be correct. In previously published data, WOX8 is expressed in the zygote, it is not specifically activated in suspensor cell.

Reviewer #1 (Remarks to the Author):

Q1. In their revised manuscript, Zhou et al. have corrected their title/abstract, included more background on previous attempts to profile apical cell and basal cell transcriptomes, and have also provided additional data and controls to confirm that their resource is of high-quality. The writing is much improved, and as I mentioned in my original review, this appears to be a high-quality dataset that should help elucidate the molecular pathways for these initial cell lineages. And although I appreciate the authors addressing most of my points, my major criticism (#1) was that describing the different amounts of lncRNAs, alternative splicing and transcript levels in apical and basal cell lineages does not address the how these cell lineages become different. I realize that such functional studies would require more experiments, but this major point was not addressed and I expect a study to have functional data to be published in a journal like Nature Communications.

Reply:

Apical and basal cell lineage specification is a crucial developmental event in plant embryogenesis. However, the molecular mechanisms regulating cell lineage specification are still largely unknown, mainly because of technical difficulties in isolating apical cell and basal cell and few available gene expression data of apical and basal cell lineages of early embryos. By overcoming technical limitations, we successfully separated apical cell and basal cell and their descendants for transcriptome analysis in the present study. Thus, our present manuscript is the first study for the transcriptome of apical cell and basal cell and transcriptome dynamic during cell lineage specification in early embryogenesis, which will provide very valuable resources for scientists in the field of plant development. In some way, it is necessary for following gene function analysis.

Yes, we agree that functional analysis of cell lineage-specific genes is critical for the understanding the molecular mechanism of cell lineage specification. Obviously, it is really a big project, not a matter of single gene analysis. Therefore, the project will take years and generations of investigators to complete. Indeed, we are carrying out functional analysis of some related genes. At present, we try to publish this important transcriptome analysis results. We wish to share these useful clues with all readers from different groups to push forward the investigations of the molecular mechanism regulating cell fate specification during embryogenesis. Based on our data, readers can easily find interesting genes to work with and are no longer to be hindered by technical difficulties in isolating the single apical and basal cells. We believe this will be welcome by all the researchers in this field.

Reviewer #2 (Remarks to the Author):

Q1. I appreciate the efforts the authors made in response to my questions and minor concerns. Specially, they provided information on additional markers analysis to strengthen their data validation. However, given a decent number of prior examples, it is not at all surprising that the apical and basal cells have distinct transcriptomes. While I applaud the effort that went into this work and the resource it provides to a specialized community, it remains a description of transcriptomes with some technical validation, without any hint of its genetic regulation or functional implications. Furthermore, there still is a lack of substantial evidence to support the speculations about the potential functions of TF, lncRNA, and alternative splicing in cell fate specification during plant embryogenesis, which should have been the highlight of the manuscript. Even though lncRNA and alternative splicing are very important for animal embryogenesis as explained by authors, it does not necessarily mean that they will have important functions in cell specification during plant embryogenesis. Based on the above, I still regard this too preliminary to be published as a standalone paper, rather than a resource.

Reply:

Apical and basal cell lineage specification during early proembryos is an important topic in the field of plant development. However, we almost know nothing about when and how the apical and basal cell lineage become distinct, not matter to mention the molecular mechanisms regulating cell lineage specification. Is it due to differential portioning of transcripts during zygote division or due to *de novo* transcription in specific cell lineage? Obviously, to answer these critical questions transcriptome analysis is necessary and function analysis of few genes is inadequate. However, it remains impossible to perform transcriptome analysis of single apical and basal cells mainly because of technical difficulties in isolating apical cell and basal cell. We have made great efforts to establish the techniques and made the analysis possible. This work, for the first time, provides a valuable lineage-specific transcriptome resource to elucidate the molecular pathways for divergence of apical and basal cell lineages at genome-wide scale. More importantly, we reveal that both selective deletion of inherited transcripts and specific *de novo* transcription contribute to the establishment of cell lineage-specific pathways for cell fate specification. Especially, we found that embryo-related pathways have been specifically activated in apical cell lineage since 1-cell embryo stage, this finding challenges our current opinion on the initiation of embryogenesis. We believe these findings provide valuable insight in understanding molecular mechanism regulating cell lineage specification as a standalone

transcriptome analysis work.

We agree that functional analysis of cell lineage-specific genes is also important but it is indeed a big project, which is worthy to be studied in the further work and will take years to complete. At present, we wish to publish this important transcriptome analysis results and share these useful clues with all readers from different groups to push forward the investigations of the molecular mechanism regulating cell fate specification during embryogenesis.

Minor point:

Q1. The information about WOX8 expression in line 353 does not appear to be correct. In previously published data, WOX8 is expressed in the zygote, it is not specifically activated in suspensor cell.

Reply: Thanks for your careful reading. The words “and WOX8 in Arabidopsis” and related reference have been removed in the discussion section (Line 350-351).